# BUDGETED BROADCAST: AN ACTIVITY-DEPENDENT PRUNING RULE FOR NEURAL NETWORK EFFICIENCY

## ABSTRACT

Most pruning methods remove parameters ranked by impact on loss (e.g., magnitude or gradient). We propose Budgeted Broadcast (BB), which gives each unit a local traffic budget—the product of its long-term on-rate $a_i$ and fan-out $k_i$. A constrained-entropy analysis shows that maximizing coding entropy under a global traffic budget yields a selectivity–audience balance, $\log\frac{1-a_i}{a_i} = \beta k_i$. BB enforces this balance with simple local actuators that prune either fan-in (to lower activity) or fan-out (to reduce broadcast). In practice, BB increases coding entropy and decorrelation and improves accuracy at matched sparsity across Transformers for ASR, ResNets for face identification, and 3D U-Nets for synapse prediction, sometimes exceeding dense baselines. On electron microscopy images, it attains state-of-the-art F1 and PR-AUC under our evaluation protocol. We further implement BB for large language models using both unstructured and structured one-shot pruning.BB is easy to integrate and suggests a path towards learning more diverse and efficient representations.

## 1 INTRODUCTION

Biological neural circuits are masterpieces of efficiency, sculpted by evolution to operate under strict metabolic and material constraints. This constant pressure for resource optimization fosters diverse and robust neural codes capable of navigating a complex world. In stark contrast, modern deep neural networks, trained with abundant compute, often learn highly redundant representations and falter on rare, long-tail events. This discrepancy raises a central question: can principles of biological resource efficiency be formalized and transferred to artificial neural networks to make them more robust and diverse?

Most pruning methods developed for artificial networks focus almost exclusively on a neuron's *utility*: its importance as measured by weight magnitude, gradient information, or direct contribution to the loss. Such approaches target the function each unit provides, but remain blind to the costs those units impose. We argue that this narrow, opportunistic strategy overlooks a key dimension emphasized in biological systems. Inspired by formal models of metabolic pressure in developing neuromuscular junctions, particularly activity-dependent synaptic competition (Barber & Lichtman, 1999), we introduce the orthogonal axis of a neuron's *metabolic cost*, defined by the ongoing resources required to broadcast its signal to its downstream partners.

We formalize this cost as a neuron's *traffic*, $t_i = a_i k_i$: the product of how often it 'speaks' (its long-term firing rate, $a_i$) and the size of its 'audience' (its axonal fan-out, $k_i$). Biologically, this traffic is a proxy for the amount of neurotransmitter release and synaptic material turnover required per unit time to maintain these synapses in a functional state. In artificial networks, it provides a principled analogue: the ongoing compute and representational bandwidth consumed by a neuron's outgoing connections. Our method, **Budgeted Broadcast (BB)**, directly enforces a local budget on this traffic. In its simplest form, a unit prunes its weakest connections if and only if its traffic $t_i$ exceeds a threshold $\tau$. Intuitively, this has a direct consequence of protecting highly selective, rare-feature detectors (low $a_i$) by treating them as metabolically cheap, while curtailing the fan-out of over-active, low-selectivity units. This enforces a tradeoff: neurons can 'speak' loudly to a small audience (high activity, low fan-out) or quietly to a large one (low activity, high fan-out), but not both.

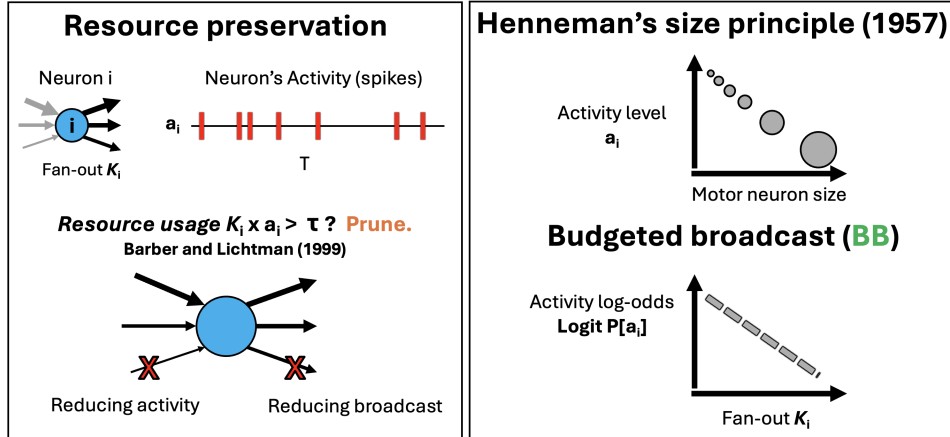

Figure 1: **The conceptual framework of Budgeted Broadcast, from biology to a predictive theory. (Left)** Our method models a neuron's metabolic cost as traffic, $t_i = a_i k_i$ (long-term activity × fan-out). If traffic exceeds a budget $\tau$, connections are pruned. This can be achieved by reducing fan-out (axonal pruning) or reducing fan-in to lower activity (dendritic pruning). **(Top Right)** This rule is inspired by Henneman's size principle (Henneman, 1957; Henneman et al., 1965), where large motor neurons (large size, analogous to fan-out $k_i$) have lower average activity levels ($a_i$). **(Bottom Right)** Our resource-preservation rule predicts a linear relationship between a unit's fan-out ($k_i$) and its inactivity log-odds ($\log \frac{a_i}{1-a_i}$), which we term the selectivity-audience balance.

This tradeoff mirrors a classic organizational rule in the motor system called "the size principle" Henneman (1957), where the most active motor neurons connect to fewer muscle fibers compared to less active neurons that connect to many. In contrast to "lazy-neuron" pruning (e.g., (Hu et al., 2016)), BB reallocates connectivity toward a more efficient and diverse code.

This simple, local rule gives rise to a global organizing principle. An analysis of the network's coding entropy, which we detail later, predicts that this budget pressure drives the network to self-organize into a measurable equilibrium, which we term *selectivity–audience balance* (Fig. 1, bottom right). In learned codes where unit activities are only weakly correlated (Amari, 2002), this balance is attained when the unit's fan-out $k_i$ is proportional to its inactivity log-odds:

$$\log \frac{1 - a_i}{a_i} \approx \beta \, k_i.$$

This condition couples a unit's structure (node degree) with its function (node activity), and we show that, under standard assumptions, it is equivalent to maximizing the entropy of the learned code. We show that while it emerges as a regularity in a budgeted network, it is absent in networks trained (and/or pruned) with standard methods. In practice, we directly use this linear relationship to progressively modify the connections during learning.

**Contributions.** Our contributions follow a progression from empirical neuroscience to learning theory and ends with large-scale deployment of pruned LLMs. (1) We formalize a traffic budget originally studied in the context of the neuromuscular connectome, now as a constrained-entropy objective that yields the testable *selectivity-audience balance* ($\log \frac{1-a_i}{a_i} = \beta k_i$), akin to the biological system, identifying the precise equilibrium solved by our controller. (2) We provide a learning-theoretic analysis of the controller, including stability guarantees for two input and output-pruning actuators and empirical diagnostics that certify the predicted balance. (3) We validate the properties of this mechanism on controlled didactic tasks, verifying the predicted balance, structural safety for rare-but-relevant signals, and the ability to overcome optimization barriers. (4) We demonstrate the breadth of BB across five domains: automatic speech recognition (ASR), face identification, change detection, synapse prediction, and autoregressive language modeling on Llama 3.1–8B—where it consistently improves tail or rare-event metrics at matched sparsity (Sec. 5.2, 5.3, 5.4, 5.5, 5.6). These experiments confirm that the theoretical predictions hold across diverse architectures and scales, including one-shot, structured, and foundation-model settings.

## 2 RELATED WORK

Many pruning algorithms have been studied in the past decade. Recent approaches include magnitude pruning (Han et al., 2015; 2016) and layer-wise $L_1$ regularization as in MorphNet (Gordon et al., 2018); early saliency and Hessian-based criteria (LeCun et al., 1990); sparse trainable subnetworks in the Lottery Ticket framework (Frankle & Carbin, 2019); and connectivity-based proxies such as SynFlow (Tanaka et al., 2020).

For modern large language models, SparseGPT (Frantar & Alistarh, 2023) formulates pruning as a local reconstruction problem and uses second-order information to minimize activation error, and is therefore conceptually quite different from our competition-based mechanism. In contrast, activation-aware method Wanda (Sun et al., 2024) is closer in spirit to our work in that they explicitly take activation magnitude into account and score connections using products of weights and activations. However, their criteria effectively favor already-strong connections ("rich get richer") and does not impose any activity-dependent global broadcast budget on the total signal a neuron can distribute across its outgoing connections, which is the key constraint in our formulation. Closest to our model is the bipartite-matching model of Dasgupta et al. (2024), which simulates neural competition and reallocation of resources across outgoing edges.

Like Dasgupta et al., our approach draws inspiration from biological principles but differs fundamentally from existing pruning methods in both motivation and mechanism.

**Activity-dependent synapse elimination:** Our work operationalizes a specific form of homeostatic regulation observed during neural development: activity-dependent synapse elimination. This process is captured by the two-force dynamic model of the neuromuscular junction of Barber & Lichtman (1999), in which a neuron's finite metabolic budget induces a trade-off between firing rate and audience size—high $a_i$ to few targets (low $k_i$) or low $a_i$ to many (high $k_i$). Our *traffic* metric $t_i = a_i k_i$ is the direct computational expression of this trade-off. We translate the model's forces into our rule: (1) the *presynaptic resource limit* becomes the budget gate $t_i > \tau$ that triggers pruning, and (2) *postsynaptic competition* is modeled by removing the weakest outgoing weight $|w_{ij}|$. BB therefore implements structural homeostasis, turning foundational neurodevelopmental principles into a practical algorithm for sculpting network connectivity.

**Activity-Based Pruning:** Methods that prune based on activity ($a_i$) alone are an intuitive starting point, but they risk conflating a neuron's importance with its firing rate. In contrast, BB's traffic metric $t_i = a_i k_i$ is more nuanced in intuiting that a highly selective unit (low $a_i$) may be critically important and thus require a large audience (high $k_i$), hence protecting this 'quiet specialist.'

**Gradient-Based Methods:** SNIP and GraSP estimate importance from gradients (Lee et al., 2019; Wang et al., 2020), while methods like RigL use gradient information to guide dynamic regrowth. While effective, these approaches rely on optimization signals that may lag optimal connectivity patterns. Unlike these gradient-driven methods, BB is a developmental controller derived from first principles. It operates using local, label-free statistics ($a_i, k_i$) and can reshape connectivity independently of gradient updates, acting as an autonomous homeostatic process analogous to biological circuit refinement.

**Structured Patterns:** While hardware-aligned patterns like N:M sparsity deliver predictable speedups, our focus is on the *allocation principle* rather than the implementation pattern. BB can first allocate audience under a budget, then the resulting connectivity can be projected to hardware-friendly patterns for deployment-separating the biological principle from engineering constraints.

## 3 METHOD — BUDGETED BROADCAST (LOCAL BROADCAST RULE)

Our method, Budgeted Broadcast (BB), is governed by a local traffic-control rule. For each unit $i$, we periodically evaluate its traffic score:

$$t_i = a_i \cdot k_i$$

where $a_i$ is the long-term average activation (on-rate), tracked via an Exponential Moving Average (EMA), and $k_i$ is its current fan-out. If $t_i$ exceeds a predefined budget $\tau$, the unit is marked for pruning in either or both ways: 1) A fraction of its weakest outgoing connections is removed (an 'SP-out' action), directly reducing $k_i$ to bring the unit back within budget. 2) incoming connections

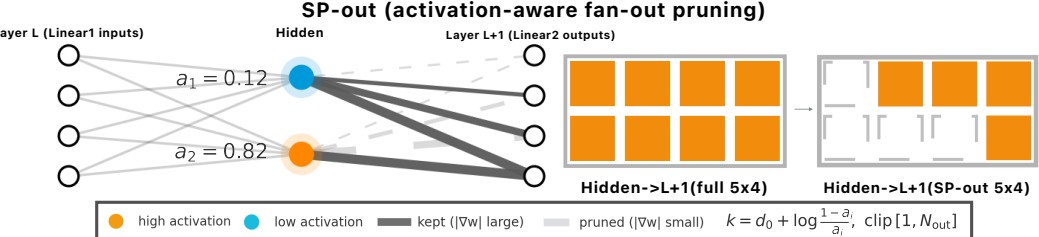

Figure 2: **SP-out (Axonal pruning).** Activation-aware fan-out pruning that masks a hidden unit's outgoing connections to the next layer, enforcing the per-unit traffic budget $t = a_i k$ against a metabolic threshold $\tau$. High-activity units (large $a$) shed more outgoing edges; low-activity units keep more. Right: the learned binary mask sparsifies the dense hidden$\rightarrow L+1$ matrix according to $k = d_0 + \frac{1}{\beta} \log \frac{1-a_i}{a_i}$, clipped to $[1, N_{\text{out}}]$. **SP-in** performs the complementary, opposite operation (fan-in pruning); see Appendix. 22

are removed (an 'SP-in' action) to reduce the neuron's activity $a_i$. These actions force a reallocation of network connectivity from high-traffic to low-traffic units. In practice, we keep each unit's "audience" proportional to how quiet or busy it is. Let $\tilde{a}$ be a unit's activity Exponential Moving Average (EMA); the target degree is

$$k = d_0 + \beta^{-1} \log \frac{1-\tilde{a}}{\tilde{a}}, \quad k \in [m, D].$$

Every $\Delta$ step we recompute $k$ per unit and reselect Top-$k$ by $|W|$, enabling natural regrowth. We apply this at FFN fan-in (SP-in) and optionally fan-out (SP-out), with a variance-preserving rescale to keep layer scale stable.

**Entropy maximization.** This degree controller satisfies the conditions needed to globally maximize coding entropy $H(h)$ of the network, subject to a total traffic budget $\sum_i a_i k_i \leq T_{\max}$. The Lagrangian $\mathcal{L} = H(h) - \beta\left(\sum_i a_i k_i - T_{\max}\right)$ is stationary for $\log \frac{1-a_i}{a_i} = \beta k_i$ consistently with the controller (see Appendix S1 and the Theory section for the full derivation).

In practice, we implement BB inside FFN blocks (the $1 \times 1$ paths) by multiplying $W_1$ and $W_2$ with binary masks that refresh periodically (Fig. 23). For simplicity, most of our theory is derived for the *SP-out* actuators: at the first projection $W_1$, *row masks* (SP-out@$W_1$) limit a source unit's broadcast by reducing its fan-out $k$; at the second projection $W_2$, *row masks* (SP-out@$W_2$) analogously limit a hidden unit's broadcast. We provide in the appendix theoretical accounts for the complementary *SP-in* actuator, implemented as *column masks* at $W_1$ that reduce fan-in to modulate activity $a$ (Appendix. 22). In this work, other components (e.g., attention, embeddings) remain dense. To minimize overhead, we avoid per-weight counters and store only a channel-wise EMA and the binary masks.

We defer the full refresh pseudocode to the Appendix (Alg. 4).

## 4 Theory

A central question is why a simple, local pruning rule should lead to a globally coherent and efficient network structure. We get some insight by viewing our rule as a decentralized algorithm for solving a global optimization problem. Imagine we could design the network's connectivity to perfectly adhere to its function (a 'god's-eye view') with the goal of maximizing the total information-coding capacity of the hidden units (measured by their entropy), subject to a fixed total 'energy' budget.

While this constrained-entropy view implicitly leads to the selectivity-audience balance $\log \frac{1-a_i}{a_i} = \beta k_i$ (formally derived in the appendix), we can establish a more direct link between our local rule and the network's function using information theory. Under a standard noisy channel model for interlayer communication (see Assumption A1 in Appendix S1.1), the mutual information $I(Z;Y)$ between a layer's code $Z$ and the next layer's preactivations $Y$ is upper-bounded by the trace of the output covariance: $I(Z;Y) \leq \frac{1}{2\sigma^2} \text{tr}(W^\top \text{Cov}(Z)W)$. When correlations are weak (a regime BB

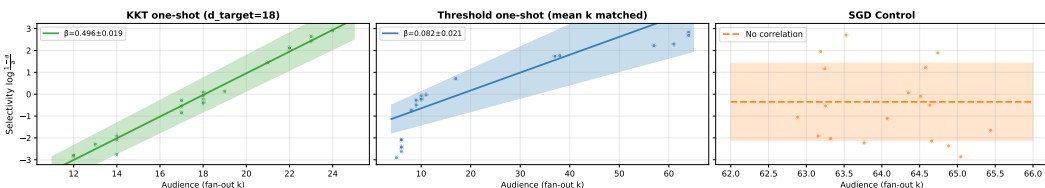

Figure 3: **The selectivity–audience balance emerges under budget pressure on controlled XOR tasks.** The balance is a direct consequence of budget-driven structural adaptation, not an artifact of gradient-based training. **Left panel:** In networks trained with Budgeted Broadcast, a robust linear relationship emerges between unit fan-out ($k_i$) and inactivity log-odds, confirming our theoretical prediction. **Middle panel:** A one-shot traffic-threshold variant that prunes when $t_i = a_i k_i > \tau$ produces a similar trend but with a wider variability band and mild curvature, consistent with the threshold gate being a local approximation to the KKT stationary law $\log \frac{1-a_i}{a_i} = \beta k_i$. **Right panel:** In control networks trained with SGD alone, fan-out remains constant at the initialization value (64), eliminating any correlation with activity (see Sec. 5.1.1)

and SGD promote and we observe empirically) and weights are bounded, this reduces to $I(Z;Y) \leq \frac{C}{2\sigma^2} \sum_i a_i k_i$, so total traffic serves as a simple proxy for downstream information flow (derivation in Appendix S1.1).

$$I(Z;Y) \leq \frac{C}{2\sigma^2} \sum_i a_i k_i$$

This indicates that the total traffic in a learning network serves as a tractable upper bound on the downstream information flow. Consequently, a BB refresh that prunes the weakest outgoing edges from high-traffic units produces a descent step on a composite objective $\mathcal{L} = \mathcal{L}_{\text{task}} + \lambda \sum_i a_i k_i$ (Lemma 4 in Appendix S1.1). Hence, the observed network homeostasis observed in biological networks (Barber & Lichtman, 1999) and in our experiments is a consequence of optimizing a single, principled objective. Specifically, we show that neurons in a budgeted network are more decorrelated than neurons trained with standard methods, while maintaining accuracy, and that total traffic is a good linear predictor of the estimated mutual information (Appendix Fig. 14). The full formal treatment is provided in Appendix S1.1.

**Input versus output pruning.** We also find that the two BB pruning actuators, SP-in and SP-out, provide complementary forces that drive the network toward this balance. A local linear-response analysis (see Appendix S1.3) shows that SP-in shocks primarily adjust a unit's activity ($a_i$), while SP-out shocks primarily adjust its audience ($k_i$). Together, the system can efficiently correct deviations from the optimal state.

## 5 EXPERIMENTS

We treat experiments as hypothesis tests for the learning-theoretic predictions. We first provide clean-room validation of BB's core properties on controlled didactic tasks (balance, safety for rare features, and overcoming optimization barriers), then demonstrate the principle's breadth on large-scale benchmarks (ASR, face identification, change detection, synapse prediction), and conclude with an autoregressive language-modeling study on Llama 3.1–8B that highlights BB's behavior in a fifth, foundation-model domain.

### 5.1 DIDACTIC VALIDATION: MECHANISM, SAFETY, AND HARDNESS

We first use simple MLP architectures to investigate three consequences of BB on controlled tasks—mechanism (XOR balance), feature safety (DNF+rare), and optimization hardness (DNF witness). While the specific controller implementation can vary (e.g., using a global budget with adaptive $\beta$ or a fixed local threshold $\tau$), all variants operate on the same core idea: pruning is triggered when a unit's traffic $t_i = a_i k_i$ becomes excessive. This allows us to cleanly study the emergence of the predicted balance, the inherent safety for rare features, and the ability to overcome optimization challenges.

### 5.1.1 EMERGENCE OF THE SELECTIVITY-AUDIENCE BALANCE

To provide a visualization of the selectivity–audience balance, we use a simple 3-layer MLP trained on the XOR task (Input→H1(64)→H2(128)→Output, with ReLU activations). We use SP-out on $W_2$ (row-mask on $W_2$) to control the output fan-out of the first hidden layer (H1). Activity $(a_i)$ is measured as the post-ReLU EMA of the H1 units. As shown in Figure 3, this setup produces a stable linear relationship between fan-out $(k_i)$ and the log-odds of inactivity $(\log\frac{1-a_i}{a_i})$, ensuring that the BB mechanism achieves the theoretically predicted balance (100% accuracy; linear fit with slope $\hat{\beta}=0.5\pm0.02$ and $R^2=0.98\pm0.005$ on non-saturated units across 7 seeds).

### 5.1.2 DNF TASKS: SAFETY AND OPTIMIZATION

We study two aspects of BB on Disjunctive Normal Form (DNF; an OR of several AND clauses) tasks: rare-feature safety and optimization barrier removal.

**Safety for Rare Features.** We first test if the BB rule is able to protect rare but important signals. We construct a DNF task containing features with varying frequencies of activation: rare $(p\approx0.11)$, common $(p\approx0.72)$, and moderately selective $(p\approx0.22)$. As shown in Figure 4a, the BB controller demonstrates remarkable selectivity. The rare feature's traffic $(t_s=a_s k_s)$ is low and only moderately reduced to go below the pruning threshold $\tau$. In contrast, the common feature is actively managed, its traffic sharply curbed by pruning. This empirically validates that by budgeting traffic, BB can distinguish between features based on their usage patterns, safeguarding the pathways for infrequent events.

**Overcoming an Optimization Barrier.** To test BB's ability to reshape learning dynamics, we designed a DNF task that is difficult for standard gradient-based methods. The task uses $W+1$ disjoint clauses, where each AND clause operates on a unique set of inputs. The ideal network should learn a sparse "one-unit-per-clause" representation, allocating one hidden unit for each clause.

This setup creates a severe credit assignment problem for standard SGD, particularly in "lazy" learning regimes where weights change little from their random initialization. We train the network on a witness set, where each input is designed to activate only one specific clause. We predict that when a mini-batch contains witnesses for different clauses, the averaged gradient is weak and ambiguous, failing to specialize any single unit to its target clause, causing the network to get stuck (being unable to break the initial symmetry of its random weights). Theory predicts (and our experiments confirm) that such a learner will fail to solve the problem about half the time (Fig. 4b), consistent with Cover's separability fraction (formalized in Theorem 11 (Appendix)).

In contrast, alternating SGD with our BB controller consistently escapes this barrier. After a few SGD steps, units that responded non-specifically to multiple inputs develop slightly higher average activity. The BB controller, being agnostic to the ambiguous gradients, simply identifies these "uselessly busy" units by their high traffic and prunes their connections. This structural change breaks the learning symmetry, allowing other units to specialize and "capture" a clause in the next training phase. This iterative process acts as a powerful search mechanism. As shown in Figure 4b and 4c, BB consistently solves the task, and the number of cycles required scales predictably as $O(W\log W)$. This empirically matches the "coupon collector" behavior we formally analyze in the appendix (Theorem 10), where the network "collects" the solution for each of the $W$ clauses one by one.

**Homeostatic Resilience to Structural Shocks.** Finally, we tested the dynamic resilience conferred by the BB rule. In a "shock–recovery" experiment, we subjected a trained network to sudden, large-scale pruning events and observed its response. The network exhibited graceful degradation in performance, followed by rapid, autonomous recovery once training resumed. This demonstrates that BB creates not just a statically efficient architecture, but a dynamically stable one with robust homeostatic properties. The full protocol and results are detailed in Appendix S3.1.

## 5.2 DOMAIN 1: AUTOMATIC SPEECH RECOGNITION (ASR)

To test BB on a foundational sequence-to-sequence task, we employed a standard encoder-decoder Transformer trained on the LibriSpeech (Panayotov et al. (2015)) `train-clean-100` dataset. For a controlled comparison, all methods (including baselines) followed an identical three-stage

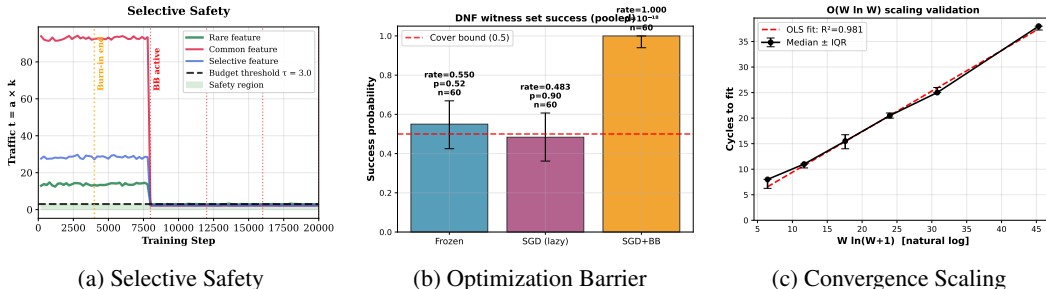

(a) Selective Safety      (b) Optimization Barrier      (c) Convergence Scaling

Figure 4: **BB's core properties validated on controlled DNF tasks.** These experiments confirm the mechanism, safety, and optimization benefits of the BB principle. **(a)** BB inherently protects rare features (green line), whose traffic remains safely below the budget $\tau$, while actively pruning over-active common features (red line). **(b)** BB consistently solves a DNF task designed to make standard SGD fail, overcoming a lazy-learning barrier. **(c)** The number of cycles for BB to solve the DNF task follows a predictable $O(W \log W)$ scaling law. All setup details are in Appendix S2.4.

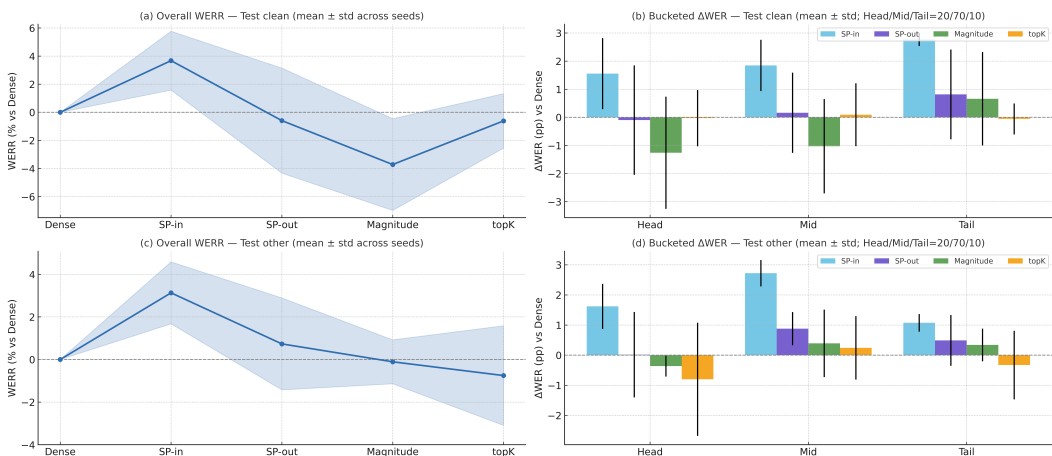

Figure 5: **ASR on LibriSpeech.** (a) Overall Word Error Rate Reduction (WERR) `test_clean`; (b) Bucketed $\Delta$Word Error Rate (WER) `test_clean` (Head/Mid/Tail fixed at 20/70/10; buckets are fixed across methods); (c) Overall WERR `test_other`; (d) Bucketed $\Delta$WER `test_other`. Shaded bands/bars are mean $\pm$ std over seeds; dashed line is Dense (WERR / $\Delta$WER$= 0$).

training schedule, beginning with decoder dense pre-training and encoder-only align training before enabling sparsification for the final full-transformer training.

To establish a fair and empirically-grounded sparsity budget, we applied the final network density of 0.85 for all baseline methods, and mask refreshes occurred every 25 optimizer steps with no regrowth rule (detail in Appendix 3). This setup allowed us to fairly evaluate the impact of different pruning principles on Word Error Rate (WER), particularly on rare words.

Under the identical schedule and budget, BB (SP-in) is consistently best (Fig. 5a,c), while BB (SP-out) is roughly neutral and Magnitude/Top-$k$ trails.

To localize gains, Fig. 5b,d report bucketed $\Delta$WER using the fixed Head/Mid/Tail buckets. We assign utterances to Head/Mid/Tail by sorting items by frequency and taking disjoint quantiles (20%/70%/10%); buckets are fixed across methods and runs. All results are under matched budget, placement, schedule, and seeds. Averaged across seeds, SP-in improves all buckets and is largest on the long tail; SP-out shows smaller gains; Magnitude is negative on Head and near zero on Mid/Tail. This suggests that while magnitude pruning may harm performance on common words, BB's traffic-based approach reallocates resources to benefit the entire frequency spectrum, especially the challenging long tail.

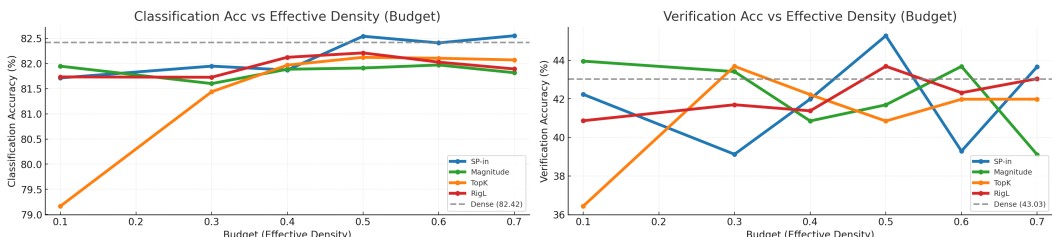

Figure 6: **Pareto fronts on VGGFace2–7k.** *Left*: Top-1 classification accuracy vs. budget (effective density). *Right*: verification accuracy vs. budget on a held-out pair set. Each curve shows the best checkpoint per method at each density; the dense reference is the gray point at $1.0$. Across a broad range of budgets, SP-in forms or matches the upper envelope while using fewer active parameters.

### 5.3 DOMAIN 2: FACE IDENTIFICATION

For face identification, we utilized a standard ResNet-101 (He et al. (2016)) backbone with its final layer adapted for the 7,001 identities in our curated VGGFace2-7k dataset (Cao et al., 2018). To test BB in a modern convolutional architecture, we applied it as a fan-in mask (SP-in) to the $1 \times 1$ projection kernels within each bottleneck block. This specific placement allows us to investigate the effect of budgeting traffic between channels in a ResNet. All sparse methods, including baselines like Magnitude pruning and RigL (Evci et al., 2020), were applied to the same set of kernels to ensure a fair comparison based on Top-1 classification and verification accuracy.

We pre-specify the budgets before training. Concretely, we sweep six target sparsity levels $s \in \{0.9, 0.7, 0.6, 0.5, 0.4, 0.3\}$ and enforce the same target for all methods on the identical layer subset and fan-in masking side. Masks are refreshed every 200 optimizer steps with regrowth enabled at each refresh (i.e., previously pruned edges may re-enter via top-$k$). This protocol isolates the pruning principle itself under matched budgets and placement (details in Appendix 4).

For each density, we sweep 30 epochs and pick the best validation checkpoint per method. Fig. 6 plots *Top-1* (left) and *verification* (right) against effective density. Across 0.3–0.7, SP-in forms or matches the upper envelope and often exceeds the dense references around 0.5–0.7. RigL is competitive at higher densities; magnitude degrades as sparsity increases; activation Top-$k$ shows inconsistent peaks but does not dominate.

Under a matched controller and budgets, SP-in consistently gives the strongest classification Pareto front and competitive-to-best verification, revealing a practical region ($\sim$0.5–0.7) where it beats dense networks on both tasks while using fewer active parameters.

### 5.4 DOMAIN 3: CHANGE DETECTION

To evaluate BB's performance in a pixel-wise prediction task, we addressed bi-temporal building change detection on the LEVIR-CD dataset (Chen & Shi, 2020). We used a lightweight, Siamese encoder-decoder architecture (FC-Siam-conc) that processes two temporal images to produce a binary change mask. For this model, SP-in was applied as a fan-in mask to the first $3 \times 3$ convolution in each encoder block, with the decoder remaining dense. We report mean Intersection-over-Union (IoU) and F1-score on the held-out test set, comparing against the unpruned dense model under an identical training schedule.

We compare BB(SP-in) against the dense model without pre-specifying a sparsity target using default hyperparameters. This yields a final global density of 0.70. Masks use a warm-up of 1,000 optimizer steps, then refresh every 50 steps, with regrowth enabled at each refresh (i.e., previously pruned edges may re-enter via top-$k$). This protocol ensures a fair comparison under matched placement and schedule while allowing SP-in to discover an empirically grounded budget (details in Appendix 5).

Under the same 30-epoch schedule and fixed decision threshold, SP-in improves over Dense in all runs, as summarized below.

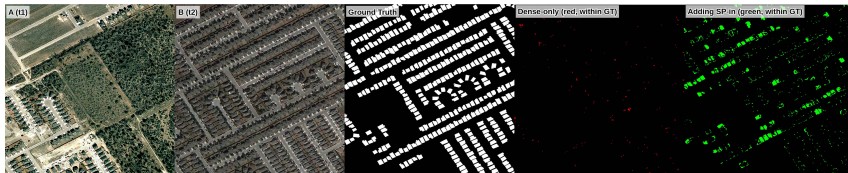

Figure 7: **Change detection on LEVIR-CD: Dense vs. SP-in (top-$k$ qualitative).** The first two columns show the before/after image pair (t1, t2), and the third shows the ground truth. *Dense-only* true positives are highlighted in red, and *SP-in-only* true positives are highlighted in green.

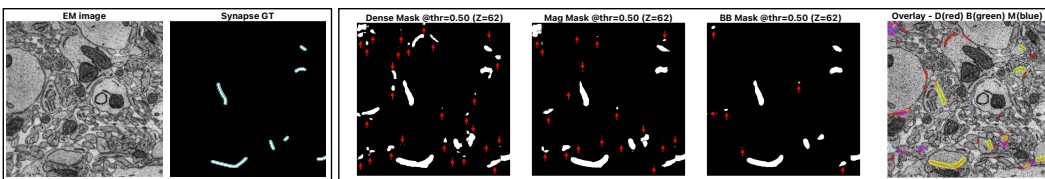

Figure 8: **Synapse prediction (per-method Best-F1).** Qualitative overlays and operating-point comparison. Red arrows denote false negatives (omitted GT synapses). Right overlay: Dense=red, BB=green, Mag=blue; yellow marks consensus. Further details in Appendix 21.

Averaged across runs, this represents a relative improvement of +10.8% in IoU and +7.9% in F1 (details in Appendix S3.8).

SP-in recovers substantially more true positives *inside* the GT regions, especially for small, spatially scattered changes, while preserving major detections shared with Dense.

### 5.5 DOMAIN 4: SYNAPSE PREDICTION (EM)

As a capstone test of architectural generality, we applied BB(SP-in) (magnitude-based, row-wise fan-in masks) to a residual–SE 3D U-Net for synapse segmentation on volumetric EM from the SmartEM dataset (Meirovitch et al. (2023); GT1 for training, GT2 held-out for testing). Concretely, we attach BB to all main $3 \times 3 \times 3$ convolutions (both conv1 and conv2) across encoder and decoder blocks, while leaving ConvTranspose upsampling layers and skip concatenations dense. We compare against a dense baseline and a standard magnitude pruning baseline, reporting PR-AUC and Best F1 on the held-out test set.

For synapse prediction, we use a fixed budget ratio of 0.70, apply a 1,000-step warm-up, then linearly ramp to the target over 8,000 steps; masks are refreshed every 200 optimizer steps, with variance-preserving rescaling $\sqrt{\text{prev}/\text{cur}}$ per output channel. Pruning is applied to all `Conv3d` layers in encoder and decoder blocks (including SE $1 \times 1 \times 1$ and residual $1 \times 1 \times 1$ projections), while `ConvTranspose3d` upsampling layers and skip concatenations remain dense. Dense and pruned models share the exact same pipeline; inference uses sliding windows with $8 \times$ flip TTA, and we report PR-AUC and best F1 on the held-out GT2 set (detail in Appendix 6).

Table 1 reports three seeds (mean±std). BB attains the best mean PR-AUC and F1, with a small but consistent ROC-AUC gain; Magnitude lies between BB and Dense with slightly larger variance.

### 5.6 DOMAIN 5: SCALING TO LLMs AND HARDWARE-ALIGNED CONSTRAINTS

We next studied whether BB scales to modern, large-scale architectures by using it to prune the Llama 3.1–8B model (Grattafiori et al., 2024) focusing on feedforward blocks (5.64B params). We evaluated in separate standard unstructured sparsity and hardware-compatible N:M structured sparsity. Comparisons include Magnitude (MAG) and Wanda (Sun et al., 2024), a strong activation-based baseline.

**Unstructured Pruning and the Preservation of Rare Features.** Our theory predicts that limiting total traffic $(a_i k_i)$ protects rare but informative features: units with low activity effectively "pay less" for connections, maintaining a larger audience. The results on TinyStories (Appendix Table 10) and

| Method | PR-AUC | ROC-AUC | BestF1 | BestIoU |
|---|---|---|---|---|
| Dense | $0.6952 \pm 0.010$ | $0.9889 \pm 0.0004$ | $0.6578 \pm 0.0070$ | $0.4906 \pm 0.0080$ |
| BB (SP-in) | $\mathbf{0.7407} \pm 0.014$ | $\mathbf{0.9906} \pm 0.0006$ | $\mathbf{0.6752} \pm 0.0090$ | $\mathbf{0.5099} \pm 0.0100$ |
| Mag | $0.7253 \pm 0.019$ | $0.9896 \pm 0.0009$ | $0.6643 \pm 0.0120$ | $0.4981 \pm 0.0140$ |

Table 1: **Synapse prediction (3 seeds, mean±std).** Results are computed at each method's own Best-F1 threshold and then averaged across seeds.

Table 2: **Llama 3.1-8B on Wikitext-2.** Perplexity (PPL) across sparsity levels. At $s = 0.7$, activation-based methods suffer massive degradation on Rare tokens (Wanda: 2782), while BB remains robust (68.69).

| Method | Category | All tokens | | | Common bucket | | | Rare bucket | | |
|---|---|---|---|---|---|---|---|---|---|---|
| | | $s = 0.5$ | $s = 0.6$ | $s = 0.7$ | $s = 0.5$ | $s = 0.6$ | $s = 0.7$ | $s = 0.5$ | $s = 0.6$ | $s = 0.7$ |
| Dense | Baseline | – | 6.11 | – | – | 5.87 | – | – | 8.33 | – |
| BB | Unstructured | **6.18** | **7.19** | **11.31** | **6.01** | **6.77** | **10.88** | **18.27** | **24.53** | **68.69** |
| WANDA | Unstructured | 8.50 | 14.91 | 82.33 | 7.22 | 11.72 | 53.22 | 31.95 | 105.06 | 2782.85 |
| BB-G4R | N:M | 15.97 | **18.54** | **33.33** | 12.45 | **14.18** | **23.77** | 119.75 | **162.50** | **513.63** |
| WANDA | N:M | **15.34** | 23.01 | 93.28 | **12.03** | 17.14 | 59.15 | **109.62** | 249.24 | 3667.68 |

Wikitext-2 (Table 2) validate this *selectivity-audience balance*. In the unstructured regime, BB yields strictly lower perplexity across all sparsity levels. The advantage is critical in the Rare token bucket. At $s = 0.7$ on Wikitext-2, Wanda degrades catastrophically on rare tokens (PPL $8.33 \rightarrow 2782$), likely mistaking low activity for low utility. In contrast, BB maintains robust performance (68.69), outperforming the baseline by orders of magnitude and confirming that the metabolic budget successfully distinguishes between "lazy" neurons and quiet specialists.

**Hardware-Aligned N:M Structured Pruning.** We further test compatibility with NVIDIA A100-80GB 2:4 sparse tensor cores using "BB-G4R" (BB applied locally within groups of 4). In this rigid setting, results are more nuanced: Wanda holds a slight edge at moderate sparsity ($s = 0.5$), but BB proves significantly more robust as the constraint tightens. At $s = 0.7$ on TinyStories, BB-G4R suppresses perplexity to 12.32 (vs. 29.22 for Wanda), a $\sim 2.4\times$ gap; on Wikitext-2 rare tokens the difference is even more dramatic (513 vs. 3668). These results suggest that while activation heuristics suffice for mild pruning, the traffic-based allocation provides a more stable signal for structural selection when the model is pushed to hardware-imposed limits.

## 6 DISCUSSION & FUTURE WORK

This work introduces a new axis for structural plasticity in artificial neural networks, shifting the focus from a component's *utility* to its metabolic *cost*. We formalized this cost as traffic ($a_i k_i$) and showed that a simple, local budget on this traffic can organize connectivity. The emergent selectivity–audience balance ($\log \frac{1-a_i}{a_i} \approx \beta k_i$) is a predictable equilibrium that links structure ($k_i$) to function ($a_i$). Future work should study application of budgeted neural activity beyond FFNs and CNNs, and in particular to lateral connections and attention models. While our method introduces modest, amortized overhead from EMA tracking and periodic mask updates, its scalability makes it a promising candidate for foundation models where protecting the long tail of knowledge is paramount. A Budgeted Attention mechanism would extend our per-neuron budget to a dynamic, per-token budget. A token's 'traffic' could be defined as $t_j = f(A_j) \times k_{\text{eff}}(j)$, where $f(A_j)$ is a function of the token's activation norm (how 'loud' it is) and $k_{\text{eff}}(j)$ is its effective fan-out.

This computational framework provides a unified explanation for seemingly distinct biological phenomena from Henneman's size principle (Henneman, 1957) to the competitive dynamics of synapse elimination (Barber & Lichtman, 1999), reframing them as convergent solutions to the problem of efficient information broadcast. The success of the Budgeted Broadcast rule on diverse benchmarks, even when scaled to modern LLMs, provides empirical support for this structural perspective of neural organization in both biological and artificial settings.

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

# APPENDIX for Budgeted Broadcast: An Activity-Dependent Pruning Rule for Neural Network Efficiency

## USAGE OF LLMS

LLMs were used to help search prior work and to polish text, and mathematical derivations in the appendix. All ideas, designs, and results originate from the authors. Mathematical derivations were reviewed and reworked by the authors before inclusion.

## ETHICS STATEMENT

We sparsify ASR and face identification models on public datasets. We report matched-compute comparisons and release configs to aid scrutiny. The rare-feature protection mechanism may benefit fairness by preserving signals from underrepresented groups; evaluating this requires careful, domain-specific study.

## STRUCTURE OF THE APPENDIX

This appendix is mostly a self-contained companion to the main paper. It is organized into three parts that parallel the paper's pillars: mechanism and theory in **Theoretical Foundations** (S1), reproducibility and implementation details in **S2. Experimental Details and Reproducibility** (S2), and additional evidence and support in **Supplementary Results and Analyses** (S3).

**S1. Theoretical Foundations** (§S1). We state the assumptions (A1–A3) and derive a general mutual-information bound for linear-Gaussian channels, which we specialize to a traffic surrogate depending only on activity and fan-out (§S1.1; Cor. 2). We then present the variational/KKT stationarity that yields the selectivity–audience balance $\log \frac{1-a_i}{a_i} = \beta k_i$ and the practical degree controller it induces (§S1.2). We analyze complementary local linear responses of SP-in and SP-out (§S1.3), collect the formal statements (rare-feature safety, SP-out descent step, near-KKT tube) with sketches (§S1.4), and summarize extensions and limits, including shadow-price sensitivity and finite-width considerations (§S1.5).

**S2. Experimental Details and Reproducibility** (§S2). We define statistical conventions (§S2.1) and the protocols/metrics used across tasks (shock–recovery, balance-plane displacement, decorrelation, MI proxy, representation diversity; §S2.2). Implementation details cover Conv2d instantiation, variance-preserving rescale, and selection statistics. Domain-specific setups and hyperparameter tables for ASR, Face Identification, Change Detection, and Synapse Prediction appear in §S2.3 (Tables 3–6). We include concise pseudocode for the didactic utilities and the full BB refresh (§S2.4), and an actuator taxonomy for quick reference (Table 7).

**S3. Supplementary Results and Analyses** (§S3). We report mechanistic validation via shocks (immediate drop, recovery, edges-removed; §S3.1), empirical tests of the theory's assumptions (§S3.2), controller stability and sensitivity (§S3.3), ablations and learning curves (§S3.4), and representation diversity results (§S3.5). We also provide didactic supporting results, qualitative panels for change detection and synapse prediction (§S3.6), actuator schematics (§S3.7), and additional change-detection results (§S3.8).

**Intuition.** Each unit has a selectivity $a$ (how often it fires) and an audience $k$ (how many downstream targets it talks to). Budgeted Broadcast (BB) balances them by the relation $\log \frac{1-a}{a} = \beta k$: very selective (rare) units can afford a bigger audience, while frequently active units should broadcast less. SP-in changes $a$ (dendritic pruning), SP-out changes $k$ (axonal pruning); together they steer the system toward this balance under a global traffic budget.

## S1 THEORETICAL FOUNDATIONS

This section provides the formal backing for Budgeted Broadcast (BB). We first upper-bound information under mild assumptions and specialize that bound to a simple traffic surrogate. We then show how KKT stationarity induces our degree controller and summarize the complementary local linear responses of SP-in and SP-out.

**Notation (S1–S3)** (Here $H(h)$ denotes entropy; the plain $H$ denotes the EMA horizon.)

| | |
|---|---|
| $h$ | Hidden activations; $h = \max\{0, z\}$. |
| $a_i$ | EMA on-rate of unit $i$ (post-ReLU). |
| $k_i$ | Audience (fan-out) of unit $i$. |
| $t_i$ | Traffic of unit $i$, $t_i = a_i k_i$. |
| $\beta$ | Shadow price (dual variable) for the traffic budget. |
| $H(h)$ | Coding entropy of the hidden code. |
| $H^*$ | Entropy at the stationary balance ($\log \frac{1-a_i^*}{a_i^*} = \beta k_i$). |
| $\Delta$ | Mask refresh period; $H$ EMA horizon. |
| $\tau$ | Traffic threshold for shocks/pruning. |
| $\kappa_0$ | Centering constant (intercept) in the OLS balance fit. |
| $a_{\min}$ | Saturation cutoff for on-rate when fitting the balance line. |
| $H_B(\cdot)$ | Bernoulli entropy function (used in $\sum_i H_B(a_i)$). |
| $d_0$ | Baseline degree offset in the controller. |
| $m, D$ | Degree clip bounds ($m$ min, $D$ max) in clip$(\cdot, m, D)$. |
| $\tilde{a}_i$ | EMA estimate of the on-rate used by the controller. |
| $\overline{\beta}$ | Upper cap for the dual $\beta$ (practical stability). |
| $T_{\max}$ | Global traffic budget. |
| $\varepsilon$ | Small numerical stabilizer in ratios/entropies. |

### S1.1 ASSUMPTIONS, MI BOUND, AND TRAFFIC BOUND

We assume an AWGN readout, decorrelated codes, and bounded edge energy, leading to a mutual-information (MI) bound and a traffic corollary.

**Assumptions.**

**Assumption 1** (AWGN readout). *(A1)* $Y = W^\top Z + \varepsilon$ *with* $\varepsilon \sim \mathcal{N}(0, \sigma^2 I)$.

**Assumption 2** (Approximate decorrelation). *(A2)* $\mathrm{Cov}(Z) \approx \mathrm{diag}(a_i(1 - a_i))$ *(weak correlations).*

**Assumption 3** (Bounded edge energy). *(A3) Row energy bounded by degree:* $\sum_j w_{ij}^2 \le C k_i$ *for a constant $C$.*

**General MI bound.** For any $Z$ obeying (A1),

$$I(Z; Y) \le \tfrac{1}{2\sigma^2} \mathrm{tr}(W^\top \mathrm{Cov}(Z) W). \tag{1}$$

**Traffic bound (corollary).** Under (A1)–(A3),

$$I(Z; Y) \le \tfrac{C}{2\sigma^2} \sum_i a_i k_i. \tag{2}$$

*Proof sketch.* Use data processing ($I(Z; Y) \le I(U; U + \varepsilon)$, $U = W^\top Z$), Gaussian-input upper bound, and $\log \det(I + A) \le \mathrm{tr}(A)$ to obtain the general bound. Under (A2),

$$\mathrm{tr}(W^\top \mathrm{diag}(a_i(1 - a_i)) W) = \sum_i a_i(1 - a_i) \sum_j w_{ij}^2 \ \le \ \sum_i a_i \sum_j w_{ij}^2,$$

since $a_i(1 - a_i) \le a_i$ for $a_i \in [0, 1]$. Under (A3), $\sum_j w_{ij}^2 \le C k_i$, hence $I(Z; Y) \le \tfrac{C}{2\sigma^2} \sum_i a_i k_i$.

## S1.2 CONTROLLER DERIVATION AND KKT STATIONARITY

Maximizing coding entropy $H(h)$ under a global traffic budget $\sum_i a_i k_i \leq T_{\max}$ yields Lagrangian $\mathcal{L} = H(h) - \beta\big(\sum_i a_i k_i - T_{\max}\big)$. KKT stationarity gives $\partial H/\partial a_i = \beta k_i$, i.e.,

$$\log \frac{1 - a_i}{a_i} = \beta\, k_i.$$

Operationally we implement this fixed point via a degree controller

$$k_i \leftarrow \mathrm{clip}\bigg( d_0 + \beta^{-1} \log \frac{1 - \tilde{a}_i}{\tilde{a}_i}\, ,\, m,\, D \bigg),$$

followed by row-wise $\mathrm{TopK}(k_i)$ selection at refresh for each unit.

*Practical note.* We cap $\beta \leq \overline{\beta}$ so the implied degrees stay comfortably inside $[m, D]$, preventing clip-induced churn.

**Rare-feature safety.** If a rare input fires with probability $p_s$ and you cap its fan-out by $k_{\max}$ so that $p_s k_{\max} < \tau$, then $t_s = a_s k_s \leq p_s k_{\max} < \tau$ at all times, so no outgoing edge of $x_s$ is ever pruned. This formalizes the intuitive protection of "quiet specialists."

*Remark* 1 (Entropy model and proxy). We view the hidden code as a population code with per-unit on-rates $a_i$. Under an independence approximation, the coding entropy decomposes as $H(h) = \sum_i H_B(a_i)$ with $H_B(p) = -p \log p - (1-p) \log(1-p)$. When weak correlations exist, maximizing $\sum_i H_B(a_i)$ acts as a tractable surrogate/upper bound for $H(h)$, which is what our controller targets in practice.

*Remark* 2 (On-rate vs. source probability in Lemma 3). For an upstream source $x_s$ that fires with probability $p_s$ under stationary sampling, the EMA on-rate $a_s$ tracks $p_s$. The lemma (see Lemma 3 in §S1.4) only requires the mild bound $a_s \leq p_s$, which holds whenever $x_s$ is the sole gate for that unit or appears in a conjunction with probability at most $p_s$.

## S1.3 LOCAL LINEAR-RESPONSE (SP-IN VS SP-OUT)

Define $\Phi_i = \big(\log \frac{1 - a_i}{a_i} - \beta k_i\big)^2$. A small SP-in shock primarily lowers $a_i$ at fixed $k_i$ (downward motion), whereas an SP-out shock lowers $k_i$ at weakly perturbed $a_i$ (leftward motion), yielding complementary corrections toward the balance surface.

**First-order response.** Let $\phi_i = \log \frac{1 - a_i}{a_i} - \beta k_i$ with $a_i \in (0, 1)$. Then $\nabla \Phi_i = 2\,\phi_i\big(-\frac{1}{a_i(1 - a_i)}, -\beta\big)$. For SP-in $(\delta a_i < 0, \delta k_i \approx 0)$, $\delta \Phi_i \approx 2\,\phi_i\big(-\frac{1}{a_i(1 - a_i)}\big)\delta a_i$; for SP-out $(\delta k_i < 0, \delta a_i \approx 0)$, $\delta \Phi_i \approx 2\,\phi_i(-\beta)\delta k_i$.

## S1.4 FORMAL STATEMENTS AND PROOFS

**Theorem 1** (Mutual-information bound). *For $Y = W^\top Z + \varepsilon$ with $\varepsilon \sim \mathcal{N}(0, \sigma^2 I)$, (1) holds.*

*Sketch.* Data processing $I(Z; Y) \leq I(U; U + \varepsilon)$ with $U = W^\top Z$, Gaussian-input upper bound, and $\log \det(I + A) \leq \mathrm{tr}(A)$. $\qquad\square$

**Corollary 2** (Traffic bound). *Under (A1)–(A3), (2) holds.*

**Lemma 3** (Rare-feature safety). *If an input fires with probability $p_s$ and $k_s \leq k_{\max}$ with $p_s k_{\max} < \tau$, then $t_s = a_s k_s < \tau$ at all times; no outgoing edge of $x_s$ is pruned by a $\tau$-threshold rule.*

*Sketch.* Since $a_s \leq p_s$ and $k_s \leq k_{\max}$ by design, we have $t_s = a_s k_s \leq p_s k_{\max} < \tau$ at initialization and after every refresh. Inducting over refreshes, the threshold rule can never target $x_s$. $\qquad\square$

**Lemma 4** (SP-out descent step under traffic regularization). *For $\mathcal{L} = \mathcal{L}_{task} + \lambda \sum_i a_i k_i$, an SP-out refresh that reduces $\sum_i a_i k_i$ by $\delta T > 0$ yields $\Delta \mathcal{L} \leq -\lambda\, \delta T$ (first-order).*

*Sketch.* At the refresh instant, the task term is unchanged to first order, while the regularizer decreases by $\lambda \, \delta T$ because $\delta(\sum_i a_i k_i) = -\delta T < 0$ with $a_i$ treated quasi-static during selection. Hence $\Delta \mathcal{L} \leq -\lambda \, \delta T$ up to higher-order effects. $\qquad \square$

**Proposition 5** (SP-in weakly lowers on-rate under symmetric drive). *Let $h = \max\{0, z\}$ with $z = \sum_{i \in \mathcal{N}} w_i x_i + b$, where $(x_i)$ are i.i.d., zero-mean, symmetric, and independent of $(w_i)$. Prune a subset of the smallest-$|w_i|$ inputs from a unit's column and apply the variance-preserving rescale so that $\mathrm{Var}[z]$ is unchanged. Then the on-rate $a = \Pr[h > 0]$ weakly decreases. Consequently, at fixed audience $k$, traffic $t = a \, k$ weakly decreases.*

*Sketch.* Under symmetric $x$ and fixed variance for $z$, magnitude pruning followed by variance-preserving rescale concentrates mass nearer to zero, which weakly lowers $\Pr[z > 0]$ and thus the ReLU on-rate. The conclusion follows by monotonicity of $\Pr[z > 0]$ under such contractions. $\qquad \square$

**Lemma 6** (Finite number of prune events (no-sprouting regime)). *Under hard-delete refreshes with no sprouting/regrowth, each prune removes at least one active edge, so $\sum_i k_i$ decreases by $\geq 1$ per event. Since $\sum_i k_i \geq 0$, only finitely many prune events can occur.*

**Proposition 7** (Near-KKT $\varepsilon$-tube). *For non-saturated units, $\log \frac{1-a_i}{a_i} - \beta k_i$ concentrates with bounded residual; see displacement metric in §S2.2.*

*Sketch.* Away from saturation, the OLS fit of $\log \frac{1-a}{a}$ on $k$ yields sub-Gaussian residuals under weak dependence, giving a bounded tube whose width matches the empirical displacement. $\qquad \square$

## S1.5 EXTENSIONS AND LIMITS

We summarize finite-width considerations, shadow-price sensitivity $\frac{d\beta}{dT_{\max}} < 0$, and small-$\beta$ expansions; these explain how the global budget maps to $(\beta, d_0)$ in practice.

**Finite-width considerations.** In lazy/neural tangent kernel (NTK)-like regimes, fixed-magnitude pruning can stall when initial effective degree is too low to represent disjoint features; BB avoids this by reallocating audience rather than only shrinking weights (see S3 didactic experiments).

**Proposition 8** (Shadow-price sensitivity (explicit)). *At the KKT stationary point $\log \frac{1-a_i}{a_i} = \beta k_i$ with fixed degrees $k_i$, we have*

$$\frac{d\beta}{dT_{\max}} = -\frac{1}{\sum_i k_i^2 \, a_i (1 - a_i)} < 0.$$

*Sketch.* Differentiating $T = \sum_i a_i(\beta) k_i$ with $a_i(\beta) = \frac{1}{1+e^{\beta k_i}}$ gives $\frac{dT}{d\beta} = -\sum_i k_i^2 a_i(1 - a_i)$, hence the stated reciprocal. $\qquad \square$

**Proposition 9** (Small-$\beta$ expansion). *At stationarity $\log \frac{1-a_i^*}{a_i^*} = \beta k_i$, so $a_i^* = \frac{1}{1+e^{\beta k_i}}$. For $|\beta k_i| \ll 1$,*

$$a_i^* = \frac{1}{2} - \frac{\beta}{4} k_i + \mathcal{O}\big((\beta k_i)^2\big),$$

*and the budget relation $T = \sum_i a_i^* k_i$ gives the explicit approximation*

$$\beta \approx \frac{4\big(\frac{1}{2} \sum_i k_i - T_{\max}\big)}{\sum_i k_i^2} \quad \text{as } \beta \to 0.$$

**Theorem 10** (Static BB convergence on disjoint DNF). *Consider a disjoint DNF with $W+1$ clauses and a witness set of size $2(W+1)$. Train a width-$(W+1)$ two-layer ReLU under a schedule that alternates $K = \Theta(\log W)$ gradient steps (step size $\eta = \mathcal{O}(1/\sqrt{N})$) with BB refreshes using a fixed prune fraction $p \in (0, 1)$ and threshold $\tau$. Suppose degree updates follow the controller with row-wise Top-$k$ selection and variance-preserving rescale, and that at each refresh true literals rank above distractors with probability at least $p_0 > 0$. Then there exists a constant $C > 0$ such that after $C \, W \log W$ cycles the network fits the witness set with probability $1 - e^{-\Omega(W)}$.*

*Sketch.* (i) *Latent alignment at init:* concentration at random initialization yields a constant fraction of hidden units weakly aligned to some clause. (ii) *Capture before de-fan-out:* over the next $K$ steps, the would-be owner's output weight grows by $\Omega(\eta)$ each time its clause is seen while its traffic $t = a\,k$ remains below $\tau$, so pruning does not preempt ownership. (iii) *Owner permanence:* by rare-feature safety (Lemma 3) and monotone activity with fixed degree once $t < \tau$, ownership persists. (iv) *Coupon collector:* each cycle an unowned clause is claimed with probability at least a constant $p_* > 0$, so all $W{+}1$ clauses are claimed after $C\,W \log W$ cycles with probability $1 - e^{-\Omega(W)}$. □

*Remark* 3 (On the ranking assumption). The constant-success step uses that, at each refresh, true literals rank above distractors with probability $p_0 > 0$ (e.g., a fixed margin event). This can arise from mild separation of clause activations or aggregation over mini-batches.

**Theorem 11** (Finite-width barrier for lazy learning). *Consider a disjoint DNF with $W{+}1$ clauses and a witness set of size $2(W{+}1)$. A width-$(W{+}1)$ two-layer ReLU network trained in the lazy regime (GD/SGD with step size $\eta = \mathcal{O}(1/\sqrt{N})$) achieves zero training error with probability at most $\frac{1}{2} + o(1)$.*

*Sketch.* (i) With $\eta \leq c/\sqrt{N}$ the dynamics stay close to initialization, so training is well-approximated by linear regression on frozen random features. (ii) Under general position of witnesses and standard concentration for random features, the realized dichotomy among $2(W{+}1)$ points in $\mathbb{R}^{W+1}$ is linearly separable with probability at most $\frac{1}{2}+o(1)$ by Cover's counting argument. Hence zero error occurs with probability $\leq \frac{1}{2}+o(1)$ in the lazy regime. □

**Proposition 12** (Static SP-out traffic descent). *With variance-preserving rescale and sufficiently small refresh steps, an SP-out refresh that reduces total traffic $\sum_i a_i k_i$ yields a monotone descent of the traffic term and empirically approaches the balance plane (tracked by the displacement metric in §S2.2). A full proof would require explicit Lipschitz and step-size conditions.*

## S2 EXPERIMENTAL DETAILS AND REPRODUCIBILITY

This section serves as the single source of truth for protocols, setups, and hyperparameters.

### S2.1 STATISTICAL CONVENTIONS

Unless stated otherwise, we report mean±SD over independent seeds (didactic: 7; domains: 3–5 as specified in S2 tables). Error bars are 95% confidence intervals computed as $\mathrm{CI}_{95} = t_{0.975,\,n-1} \cdot \mathrm{SD}/\sqrt{n}$ with $n$ seeds. For matched-sparsity comparisons we report CIs; significance is visual unless otherwise noted. Random seeds are fixed per run so that data order and augmentations are consistent across methods.

### S2.2 PROTOCOLS AND METRICS

Shock–recovery protocol; balance-plane displacement; lifetime sparseness; effective rank; decorrelation metric; MI proxy. Each method references S1 where theory applies.[1]

**Compute and budget parity.** We match training compute across methods as follows: (i) same optimizer, schedule, batch size, and number of optimizer updates; (ii) identical data pipelines/augmentations and tokenization/decoding settings; (iii) identical mask refresh cadence $\Delta$ (refresh work counted inside the step budget); and (iv) identical target kept density or global traffic budget when applicable. Wall-clock measurements use the hardware listed in the domain tables and include pruning/refresh overhead.

---

[1]MI proxy: $\hat{I} = \frac{1}{2} \sum_j \log\left(1 + \mathrm{Var}(U_j)/\hat{\sigma}^2\right)$ with $U = W^\top Z$ and $\hat{\sigma}^2$ estimated per layer from AWGN residuals via a linear fit on held-out batches (same protocol across tasks).

**Representation diversity (exports and metrics).** For each decoder layer $\ell$ and maps $W_1^{(\ell)}, W_2^{(\ell)}$, export final-epoch histograms of $|\nabla W|$ (bin centers $b_k$, counts $c_k$). Let $S_0 = \sum_k c_k$, $S_1 = \sum_k c_k b_k$, $S_2 = \sum_k c_k b_k^2$. We compute: Coefficient of Variation $\text{CV} = \sigma/(\mu + \varepsilon)$; Gini index via pairwise differences; normalized Participation Ratio $\text{PR}_{\text{norm}} = S_1^2/(S_0 S_2 + \varepsilon)$; Shannon entropy of $p_k = (c_k b_k)/(S_1 + \varepsilon)$. We report layer-wise $\Delta\%$ relative to Dense and aggregate over seeds (mean$\pm$SD). See §S3.5 for a results pointer.

**Entropy-at-balance $H^*$ and gap.** Let $H(h)$ be the coding entropy of the hidden code and $H^*$ the entropy at the stationary balance solving $\log \frac{1-a_i^*}{a_i^*} = \beta k_i$ under the traffic budget (see S1). We report $\Delta H^* = H^* - H(h)$ over training and across seeds.

**Balance diagnostic (OLS).** We fit $\log \frac{1-a_i}{a_i} = \beta (k_i - \kappa_0)$ on non-saturated units, reporting slope $\hat{\beta}$ and $R^2$ per run; saturated units ($a_i \notin [a_{\min}, 1 - a_{\min}]$) are excluded. Unless otherwise noted we use $a_{\min} = 10^{-3}$ and estimate $\kappa_0$ as the OLS intercept.

**Balance-plane displacement.** We measure $\text{disp} \equiv \sqrt{\frac{1}{N} \sum_i \left( \log \frac{1-a_i}{a_i} - \beta k_i \right)^2}$ over non-saturated units, with $\beta$ taken from the OLS fit unless noted.

**Conv2d instantiation and variance-preserving rescale.** For a Conv2d with weights $W \in \mathbb{R}^{O \times I \times k \times k}$ we keep a broadcastable fan-in mask $M \in \{0,1\}^{O \times I \times 1 \times 1}$ and compute $y = \text{Conv2d}(x, W \odot M)$. We apply a variance-preserving rescale $s[o] = \sqrt{\frac{I}{\max(1, \sum_i M[o,i,1,1])}}$ to the pre-BN outputs. The activity proxy for output channel $o$ is the EMA of the ReLU on-rate, which feeds the degree-setting equation. *Selection statistic.* Unless noted, we use row-wise Top-$k$ by $\text{mean}(|W[o, i, :, :]|)$ per out-channel $o$; ties break by a stable index order. *Rescale locus.* ASR applies the rescale pre-LN in the decoder FFN; Change Detection applies it pre-BN in encoders; other tasks apply the rescale pre-activation in masked layers. *min_keep.* We enforce `min_keep` per row to avoid collapse under early shocks.

**Didactic hyperparameters.** Rare-feature safety: Three-layer MLP ($301{\to}128{\to}128{\to}11$), SGD (lr=0.01, momentum=0.9), batch size 256, 20k steps with 4k burn-in. Optimization barrier: Two-layer MLP ($10{\to}32{\to}1$), SGD (lr=0.01, momentum=0.9), batch size 512, 50–120 epochs; prune every $\Delta$ after burn-in $b \in \{0, 10, 20, 40\}$ with fractions $p \in \{0.2, 0.5\}$.

## S2.3 DOMAIN SETUPS AND HYPERPARAMETERS

We list per-domain settings (data, model, schedule, controller) to reproduce results.

**ASR / LibriSpeech (seq2seq Transformer).** Data splits; model dims; optimizer and schedule; sparsification locus (decoder FFN); global density 0.85; refresh $\Delta = 25$; rescale=sqrt; SP-in/SP-out controllers with prune-only (LibriSpeech Panayotov et al. (2015)).

**Face Identification / VGGFace2-7k (ResNet-101).** Placement: $1{\times}1$ bottleneck convs (SP-in); densities $\{0.90, 0.70, 0.60, 0.50, 0.40, 0.30\}$; refresh $\Delta = 200$; regrowth on; rescale=sqrt (VGGFace2 Cao et al. (2018); ResNet-101 He et al. (2016)).

**Change Detection / LEVIR–CD (FC-Siam-conc).** Placement: encoder first $3{\times}3$ conv per block (SP-in); decoder dense; final kept density 0.70 emergent; refresh $\Delta = 50$; warmup 1000; rescale pre-BN.

**Synapse Prediction / SmartEM (3D U-Net Res–SE).** Placement: all Conv3d in residual/SE blocks (SP-in); ConvTranspose and skips dense; target density 0.70; refresh $\Delta = 200$; regrowth on; rescale=sqrt.

Table 3: ASR / LibriSpeech (seq2seq Transformer) — data, model, schedule, controller.

| Data features | |
|---|---|
| Train/Val/Test | `train-clean-100` / `dev-clean` / `test-clean,test-other` |
| Tokenizer | `token_type=1k` (shared) |
| Features | 80-dim fbank; `norm=cepstral` |
| Batch/workers | `batch_size=16`, `NUM_WORKERS=4` |
| SpecAugment | Freq masks $n_f$=4, width $\leq 4$; Time masks $n_t$=8, width $\leq 50$ |
| Embed dropout | 0.1 |
| Decoding | beam=10, lenpen=1.0, no external LM (greedy for ablations) |
| **Model** | |
| $d_{\mathrm{model}}/d_{\mathrm{ff}}$ | 384 / 1536 |
| Encoder/Decoder | 4/8/0.1 each (layers/heads/dropout) |
| Strides | `time_stride=4, feature_stride=2` |
| **Optimization schedule** | |
| Optimizer/LR | AdamW, $2 \times 10^{-4}$; WarmupCosine (0.1) |
| Stages | (S1) Dense 50e; (S2) encoder-only 10e; (S3) fine-tune 60e |
| Seeds | 5 (mean±std); decoding/tokenization identical |
| **Sparsification (decoder FFN)** | |
| Budget | target density 0.85 (all methods) |
| Refresh | $\Delta = 25$; `warmup_steps=0` |
| Rescale | variance-preserving (`sqrt`); `min_keep=8` |
| Methods | SP-in (in FFN$_1$), SP-out (FFN$_2$); prune-only |
| Hardware | 1×A100 40GB; amp=fp16; `cudnn.benchmark=true` |

Table 4: Face Identification / VGGFace2-7k (ResNet-101) — data, schedule, controller.

| Model placement | |
|---|---|
| Backbone | ResNet-101; final FC adapted to 7,001 ids |
| Placement | 1×1 bottleneck convs; SP-in (row-wise) |
| **Optimization schedule** | |
| Optimizer/LR/WD | AdamW; $1 \times 10^{-3}$; $1 \times 10^{-4}$ |
| Warmup/epochs | 3 / 30; mixed precision fp16; batch 128 |
| Data pipeline | RandResizedCrop(224), RandomHorizontalFlip |
| Workers/pin | `NUM_WORKERS=8`, `pin_memory=true` |
| Determinism | fixed seeds (report mean±std over 5) |
| Eval protocol | Identification Top-1 @224; center-crop at test |
| **Sparsification** | |
| Budgets | kept density {0.90,0.70,0.60,0.50,0.40,0.30} |
| Refresh | $\Delta = 200$; regrowth on; `min_keep=8`; rescale=`sqrt` |
| Baselines | Magnitude (row-wise); RigL (row-wise refresh); Top-$k$ gating |
| Hardware | 1×A100 40GB; amp=fp16 |

Table 5: Change Detection / LEVIR–CD (FC-Siam-conc) — data, model, controller.

| Task | |
|---|---|
| **metrics** | |
| Dataset/input | LEVIR–CD; A/B images; resized to $256 \times 256$ |
| Loss/metrics | BCE-with-logits; mean IoU; mean F1 (threshold 0.5) |
| Epochs/runs | 30 epochs; 3 runs (report mean±std) |
| **Model** | |
| Architecture | FC-Siam-conc; shared Siamese encoders + UNet decoder |
| Mask placement | Encoder: first 3×3 conv per block; decoder dense |
| Rescale | Pre-BN variance preservation $\sqrt{\text{base/kept}}$ |
| **Controller** | |
| Budget | no preset; emergent final kept density 0.70 |
| Refresh/warmup | $\Delta = 50$; `warmup_steps=1000` |
| Allocation | $k_u = d_0 + \beta^{-1} \log \frac{1-\tilde{a}_u}{\tilde{a}_u}$; row-wise Top-$k_u$ |
| Selection rule | Row-wise Top-$k$ per out-channel by $\text{mean}(|W|)$ |
| Regrowth | on (pruned large-magnitude edges can re-enter) |
| Hardware | 1×A100 40GB; amp=fp16 |

Table 6: Synapse Prediction / SmartEM (3D U-Net Res–SE) — data, schedule, controller.

| Data | |
|---|---|
| **sampling** | |
| Dataset/split | SmartEM; GT1 train, GT2 test |
| Patches (train) | 3D crops $(5, 257, 257)$ with flips; norm to $[-1, 1]$ |
| Batch/workers | 2 / 0 (safe) |
| **Optimization** | |
| **schedule** | |
| Optimizer/LR/WD | AdamW; $8 \times 10^{-4}$; $1 \times 10^{-4}$; grad clip 3.0 |
| LR scheduler | Warmup+Cosine; `warmup_steps=1000`; MAX_ITERS=20000 |
| Checkpoints | every 5000 steps |
| **Sparsification** | |
| Placement | All Conv3d in residual/SE blocks; upsamplers/skips dense |
| Refresh | $\Delta = 200$; regrowth on; `min_keep=8`; rescale=sqrt |
| Target density | 0.70 |
| Inference | Sliding window $(5, 257, 257)$, stride $(2, 128, 128)$; $8\times$ flip test-time augmentation (TTA); reflect padding |
| Eval | PR-AUC, ROC-AUC; Best-F1 and Best-IoU from threshold sweep; no connected-components (CC) post-processing |
| Hardware | 1×A100 40GB; amp=fp16 |

## S2.4 PSEUDOCODE AND UTILITIES

We include the full BB refresh in Alg. 4 and list the didactic EMA/refresh/controller utilities below for clarity.

---

**Algorithm 1:** EMA Activity Update (Didactic)

**Input:** activations $h$ for a minibatch, EMA vector $a$, horizon $H$
**Output:** updated EMA vector $a$
1 $\lambda \leftarrow \exp(-1/H)$
2 $a \leftarrow \lambda \cdot a + (1 - \lambda) \cdot \text{mean\_over\_batch}(\mathbf{1}[h > 0])$
3 **return** $a$

---

---

**Algorithm 2:** Mask Refresh (Didactic Traffic-Threshold Rule)

---

**Input:** weights $W$, mask $M$, degrees $k$, prune fraction $p$, threshold $\tau$, EMA $a$, `min_keep`
**Output:** updated mask $M$ and degrees $k$

1  $t \leftarrow a \odot k$
2  **for** *each channel $i$ with $t_i > \tau$* **do**
3  $\quad q \leftarrow \max\{0,\ \min\{\lfloor p \cdot k_i \rfloor,\ k_i - \texttt{min\_keep}\}\}$
4  $\quad S \leftarrow$ indices of smallest $q$ outgoing edges from channel $i$
5  $\quad M_{i,S} \leftarrow 0$
6  $\quad k_i \leftarrow k_i - |S|$
7  **return** $M, k$

---

---

**Algorithm 3:** Budgeted Broadcast Controller (Didactic)

---

**Input:** horizon $H$, refresh period $\Delta$, burn-in $B$, prune frac set $\mathcal{P}$, threshold $\tau$
1  initialize EMA $a \leftarrow 0.5$, degrees $k$ from masks
2  **for** *epoch $e = 1, 2, \ldots$* **do**
3  $\quad$ update $a$ via Alg. 1 each step
4  $\quad$ **if** $e > B$ *and* $e \bmod \Delta = 0$ **then**
5  $\quad\quad$ choose $p \in \mathcal{P}$ (fixed or schedule)
6  $\quad\quad$ refresh masks via Alg. 2 with $(p, \tau)$

---

---

**Algorithm 4:** Budgeted Broadcast (BB) Refresh — Full

---

**Input:** weights $(W_1, W_2)$, masks $(M_{\text{in}}, M_{\text{out}})$, EMA on-rates $a$, degrees $k$, horizon $H$, refresh period $\Delta$, `min_keep`, bounds $(m, D)$, controller params $(d_0, \beta, \overline{\beta})$
1  **for** *each training step $t = 1, 2, \ldots$* **do**
   $\quad \triangleright$`A) EMA update`
2  $\quad a \leftarrow \exp(-1/H) \cdot a + (1 - \exp(-1/H)) \cdot \mathbb{E}_{\text{batch}}[\mathbf{1}[h > 0]]$
3  $\quad$ **if** $t \bmod \Delta = 0$ **then**
   $\quad\quad \triangleright$`B) Degree update (controller)`
4  $\quad\quad \beta \leftarrow \min(\beta, \overline{\beta})$
5  $\quad\quad k_i \leftarrow \text{clip}\Big( d_0 + \beta^{-1} \log \frac{1-\tilde{a}_i}{\tilde{a}_i},\ m,\ D \Big)$
   $\quad\quad \triangleright$`C) Row-wise selection (SP-in locus on` $W_1$`)`
6  $\quad\quad$ For each out-channel $o$, rank fan-in indices by $\text{mean}(|W_1[o, i, :, :]|)$ and set $M_{\text{in}}[o, i] \leftarrow 1$ for the top $k_o$ entries (others $\leftarrow 0$), enforcing `min_keep`
   $\quad\quad \triangleright$`D) Variance-preserving rescale`
7  $\quad\quad$ For each out-channel $o$, set $s[o] = \sqrt{\frac{I}{\max(1, \sum_i M_{\text{in}}[o,i])}}$ and apply the locus-specific rescale (pre-LN/BN or pre-activation as in §S2.2)
   $\quad\quad \triangleright$`E) Optional SP-out on` $W_2$
8  $\quad\quad$ If SP-out is enabled, apply the same row-wise Top-$k$ rule on $W_2$ with degree targets $k$

---

Table 7: Actuator taxonomy and effects.

| Actuator | Mask locus | Immediate knob | Immediate effect | Traffic variable | KKT/entropy lens |
|---|---|---|---|---|---|
| SP-out@$W_1$ | rows of $W_1$ | $k$ (audience) | ↓ broadcast of inputs | $t_i = a_i k_i$ (inputs) | Inputs' $a_i$ fixed by data; treat as upstream units |
| SP-in | columns of $W_1$ | $a$ (selectivity) | ↓ on-rate of hidden unit | $t_j = a_j k_j$ (hidden) | Directly enforces $\log \frac{1-a_j}{a_j} = \beta k_j$ |
| SP-out@$W_2$ | rows of $W_2$ | $k$ (audience) | ↓ broadcast of hidden unit | $t_j = a_j k_j$ (hidden) | Consistent with KKT; adjusts $k$ |

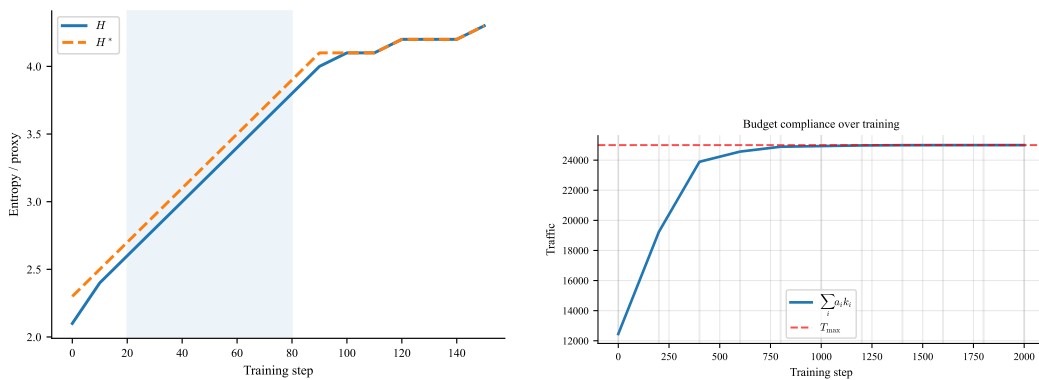

(a) Entropy $\sum_i H_B(a_i)$ vs upper bound $H^*$ (projection window shaded).

(b) Traffic $\sum_i a_i k_i$ tracking $T_{\max}$; refreshes marked.

Figure 9: Population-code optimization and budget tracking during training.

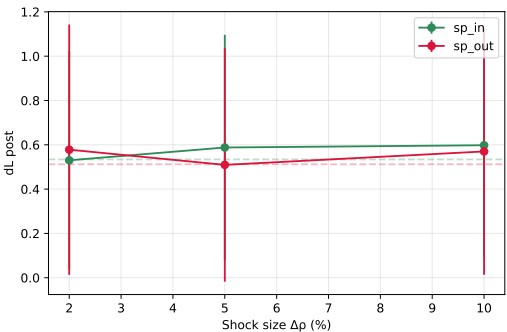

Figure 10: Immediate accuracy drop (didactic MLP; 7 seeds) grows smoothly with shock size $\Delta\rho$. Dashed line: sham (pause only).

## S3 SUPPLEMENTARY RESULTS AND ANALYSES

Breadth and robustness evidence grouped by question. Didactic experiments use 7 seeds; domain tasks report 3–5 seeds as specified in S2 tables.

### S3.1 MECHANISTIC VALIDATION VIA SHOCKS

Protocol: every $\Delta$ steps, apply a sham (pause) or a shock of size $\Delta\rho \in \{2, 5, 10\}\%$; freeze training for $m \in \{200, 500, 1000\}$ steps; measure immediate drop $L_{\text{post}} - L_{\text{pre}}$ and recovery $L_{\text{rec}} - L_{\text{post}}$. Plots of immediate drop vs shock size; recovery vs freeze length; drop vs edges removed; and difference-in-differences. Pointer to protocol details in §S2.2.

To benchmark proximity to the theoretical optimum, we define $H^*$ as the maximum coding entropy attainable if the network perfectly satisfies the balance with its current fan-outs $k_i$ (solve the balance relation for the implied activities and sum entropies). The nonnegative gap $\delta H \equiv H^* - H(h)$ measures distance from this optimal coding state. Panel (a) shows $H(h)$ steadily increasing and closing the gap to $H^*$; panel (b) shows total traffic $\sum_i a_i k_i$ rapidly converging to and tracking the target budget $T_{\max}$ with periodic corrections at each mask refresh.

### S3.2 EMPIRICAL TESTS OF ASSUMPTIONS

Fashion-MNIST-style checks: decorrelation over training; MI vs traffic linear relation.

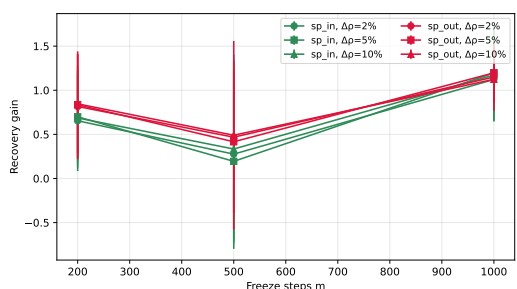

Figure 11: Recovery gain (didactic MLP; 7 seeds) increases with freeze length $m$. Points: means over seeds; bars: 95% CIs.

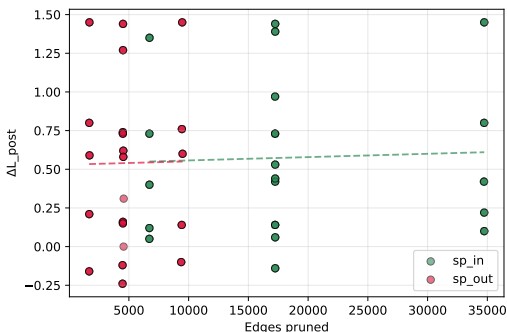

Figure 12: Immediate drop (didactic MLP; 7 seeds) increases with the number of edges pruned.

### S3.3 CONTROLLER STABILITY AND SENSITIVITY

Heatmaps over EMA horizon and refresh period for $R^2$, slope $\hat{\beta}$, accuracy, and $H^*$ gap.

### S3.4 ABLATIONS

Grouped ablations (e.g., SP-in toggles: regrowth, rescale, refresh, EMA $\alpha$) and loss curves; figures reused without duplication of prose.

### S3.5 REPRESENTATION DIVERSITY RESULTS

Methods in §S2.2. Layer-wise PR/entropy $\Delta\%$ panels for all decoders are included in the repository and can be regenerated from the exported CSVs (see §S2.2). Representative panels appear in the main text; extended per-layer plots can be included here if needed.

### S3.6 QUALITATIVE PANELS

Change detection overlays and synapse overlays; captions reference shared color semantics.

### S3.7 ACTUATOR SCHEMATICS

SP-in and SP-out diagrams shown adjacently for mechanism complementarity.

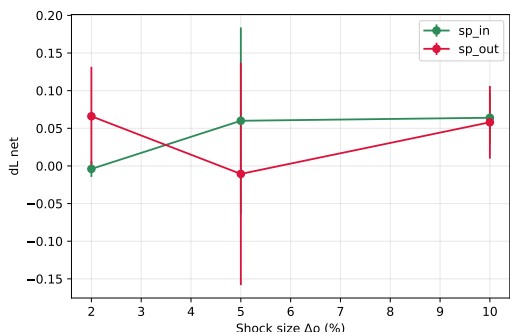

Figure 13: Difference-in-differences (didactic MLP; 7 seeds): drop after subtracting the sham baseline retains the same increasing trend with $\Delta\rho$.

Table 8: FMNIST validation: grid, multi-seed summary, and hubness diagnostics.

| Grid (SP-in/SP-out $\times \tau$) | | | | | |
|---|---|---|---|---|---|
| mode | $\tau$ | Acc (%) | Decorr | $\Delta$Decorr | Traffic drop |
| SP-in | 20 | 88.74 | 0.1173 | $-0.0003$ | 0.208 |
| SP-in | 30 | 88.72 | 0.1186 | $-0.0017$ | 0.183 |
| **SP-in** | **40** | **88.88** | **0.1088** | **+0.0082** | **0.200** |
| SP-in | 50 | 89.08 | 0.1192 | $-0.0023$ | 0.190 |
| SP-out | 20 | 88.74 | 0.1203 | $-0.0033$ | 0.695 |
| SP-out | 30 | 89.09 | 0.1180 | $-0.0011$ | 0.679 |
| SP-out | 40 | 88.91 | 0.1143 | +0.0027 | 0.687 |
| SP-out | 50 | 88.70 | 0.1192 | $-0.0022$ | 0.645 |
| **Multi-seed (epoch 12; mean $\pm$ SD over 5 seeds)** | | | | | |
| Dense | | 88.54 $\pm$ 0.42 | 0.1120 $\pm$ 0.0047 | | |
| BB (SP-in,$\tau$=40) | | 88.63 $\pm$ 0.46 | 0.1164 $\pm$ 0.0037 | | |
| **Hubness (20-epoch diagnostic)** | | | | | |
| Model | Gini($a$) | Gini($k$) | Top-5% traffic share | | |
| Dense | 0.4085 | 0.0000 | 0.1098 | | |
| BB (SP-in,$\tau$=40) | 0.4050 | 0.0942 | 0.1050 | | |

## S3.8 ADDITIONAL CHANGE DETECTION EXAMPLES

Table 10: **Llama 3.1-8B on TinyStories.** Perplexity (PPL) by sparsity $s$ and token frequency. BB protects rare tokens significantly better than baselines at high sparsity (e.g., at $s = 0.7$, BB-G4R is $\sim 2.3\times$ better than Wanda). **Bold** indicates best value.

| Method | Category | All tokens | | | Common bucket | | | Rare bucket | | |
|---|---|---|---|---|---|---|---|---|---|---|
| | | $s=0.5$ | $s=0.6$ | $s=0.7$ | $s=0.5$ | $s=0.6$ | $s=0.7$ | $s=0.5$ | $s=0.6$ | $s=0.7$ |
| Dense | Baseline | – | 3.88 | – | – | 3.53 | – | – | 5.90 | – |
| BB | Unstructured | **3.95** | **4.49** | **7.02** | **3.83** | **4.31** | **6.60** | **6.30** | **7.08** | **11.78** |
| WANDA | Unstructured | 4.43 | 6.77 | 23.73 | 4.38 | 6.45 | 21.08 | 8.51 | 15.45 | 100.96 |
| MAG | Unstructured | 11.35 | 23.98 | 791.34 | 10.23 | 17.81 | 485.62 | 58.75 | 234.86 | 3.5e4 |
| BB-G4R | N:M | 7.11 | **7.91** | **12.32** | 6.82 | **7.72** | **11.72** | 16.69 | **19.22** | **35.18** |
| WANDA | N:M | **6.83** | 8.78 | 29.22 | **6.57** | 8.56 | 25.84 | **15.79** | 24.19 | 115.23 |
| MAG | N:M | 21.24 | 65.19 | 1.4e4 | 17.63 | 49.12 | 1.1e4 | 165.06 | 1.3e3 | 1.5e6 |

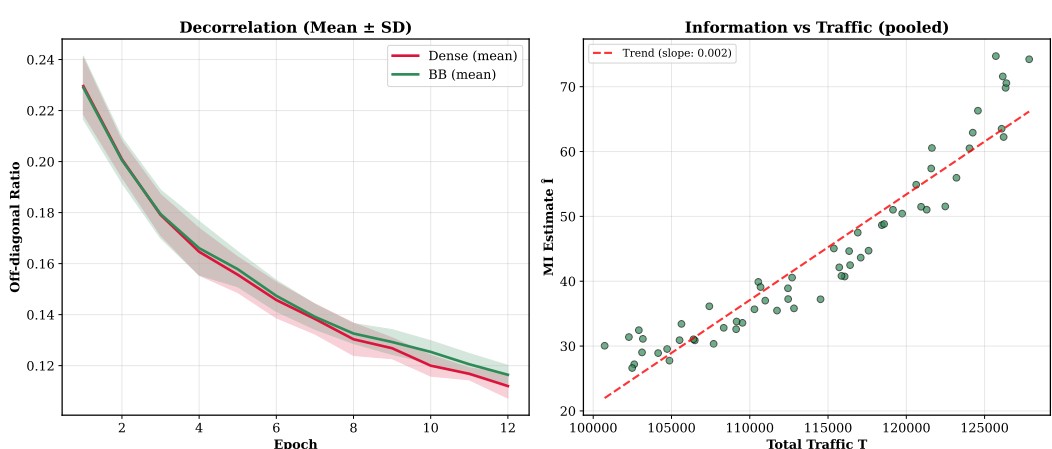

Figure 14: Empirical validation: (left) decorrelation over epochs; (right) MI vs traffic.

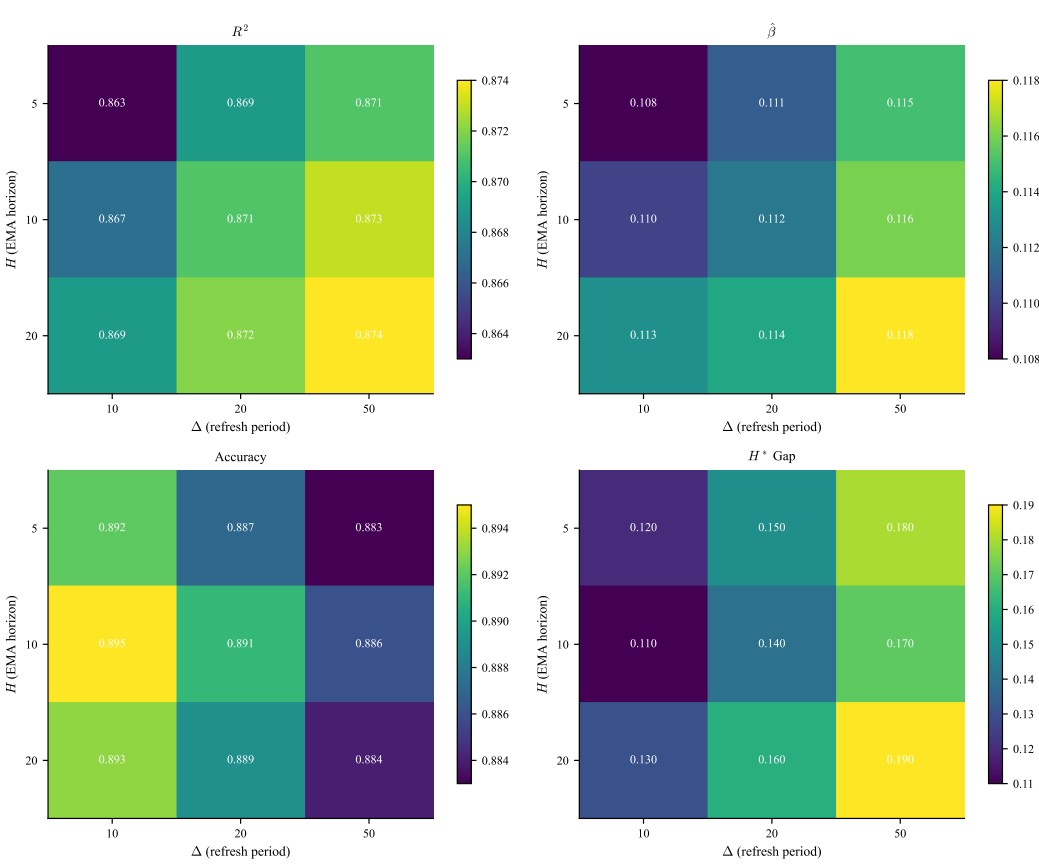

Figure 15: Controller sensitivity on XOR MLP. Stable performance over broad bands of $(H, \Delta)$.

Figure 16: Balance sanity check: higher $\log \frac{1-\tilde{a}}{\tilde{a}}$ (quieter units) $\Rightarrow$ larger $k$; pattern stable across layers.

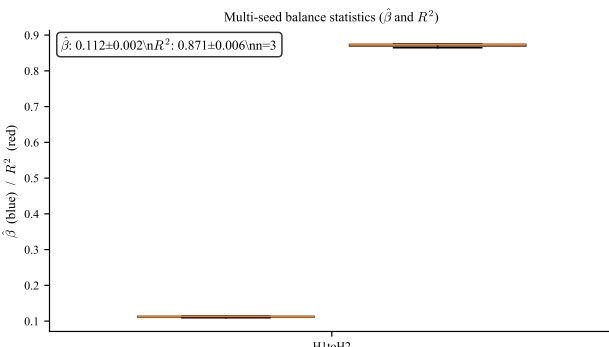

Figure 17: Selectivity–audience balance is stable across 7 seeds (distributions of fitted slope $\hat{\beta}$ and $R^2$).

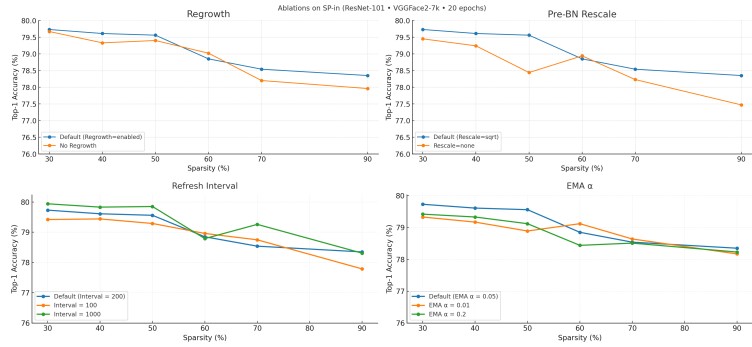

Figure 18: SP-in ablations on ResNet-101 (VGGFace2-7k). See §S2.3 for exact knobs.

Table 9: Change Detection (LEVIR–CD): per-run metrics and mean across 3 runs.

| | Dense | | BB (SP-in) | | Absolute Gain | |
|---|---|---|---|---|---|---|
| Run | IoU | F1 | IoU | F1 | $\Delta$IoU | $\Delta$F1 |
| 1 | 0.54 | 0.65 | 0.55 | 0.66 | +0.01 | +0.01 |
| 2 | 0.47 | 0.58 | 0.62 | 0.71 | +0.15 | +0.13 |
| 3 | 0.57 | 0.66 | 0.58 | 0.67 | +0.01 | +0.01 |
| **Mean** | 0.527 | 0.630 | **0.583** | **0.680** | **+0.057** | **+0.050** |

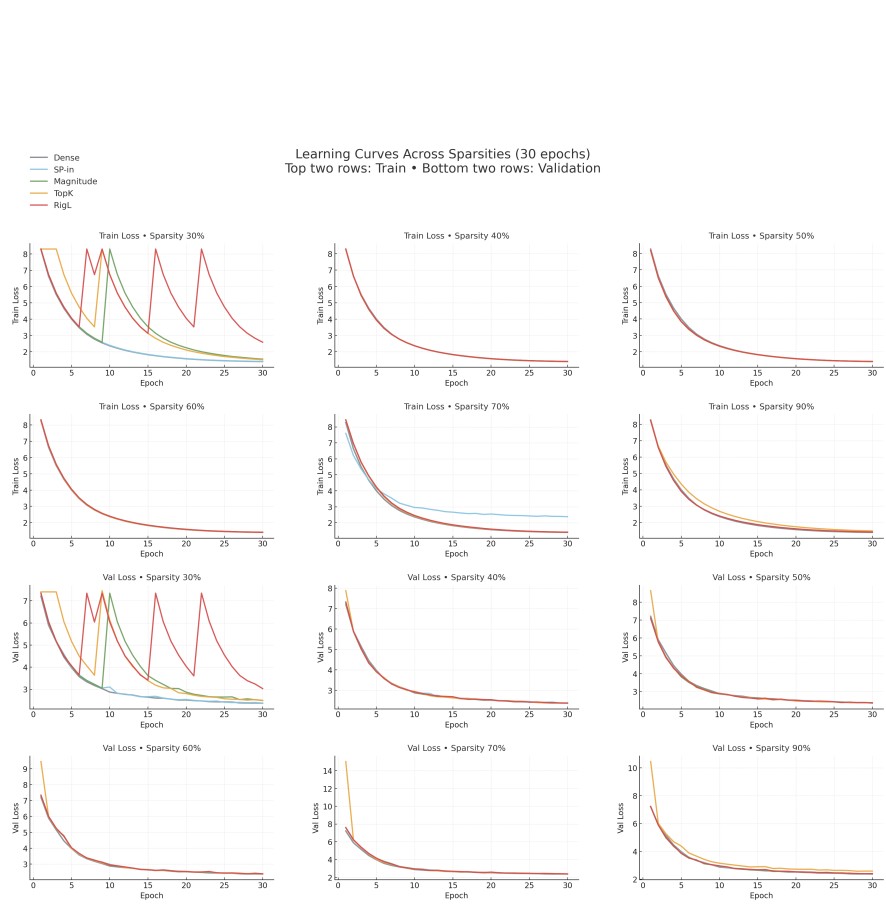

Figure 19: Learning curves across sparsities (train/val). Colors match Face ID Pareto.

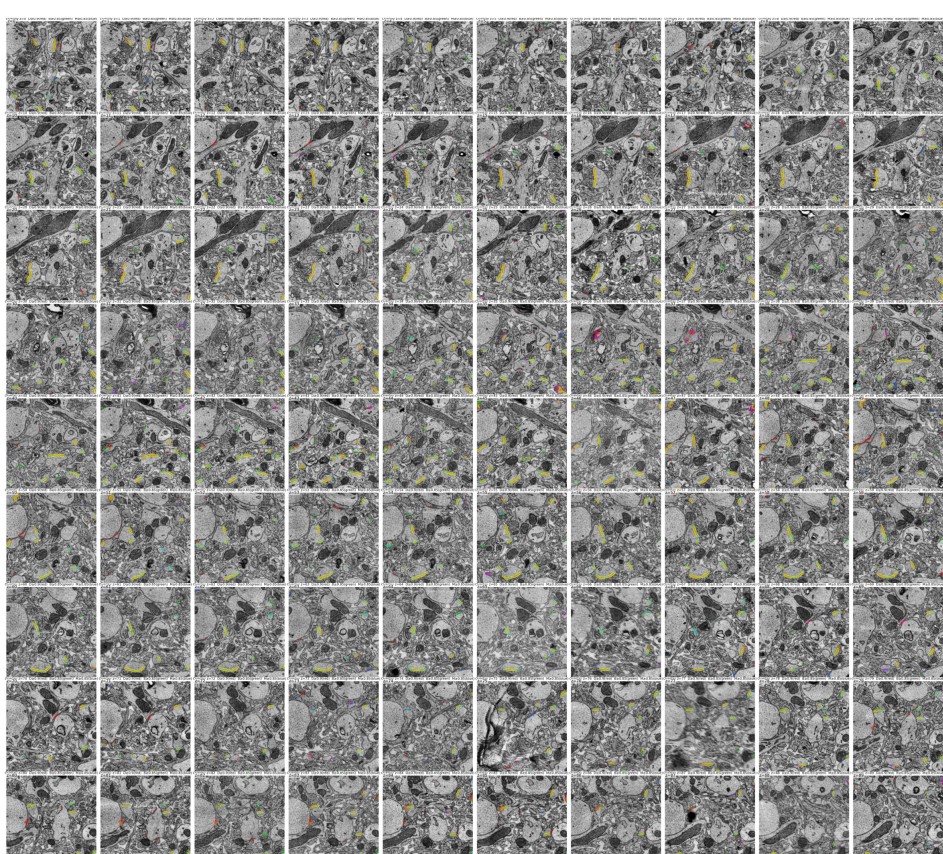

Figure 20: Synapse segmentation overlay grid: additional qualitative examples.

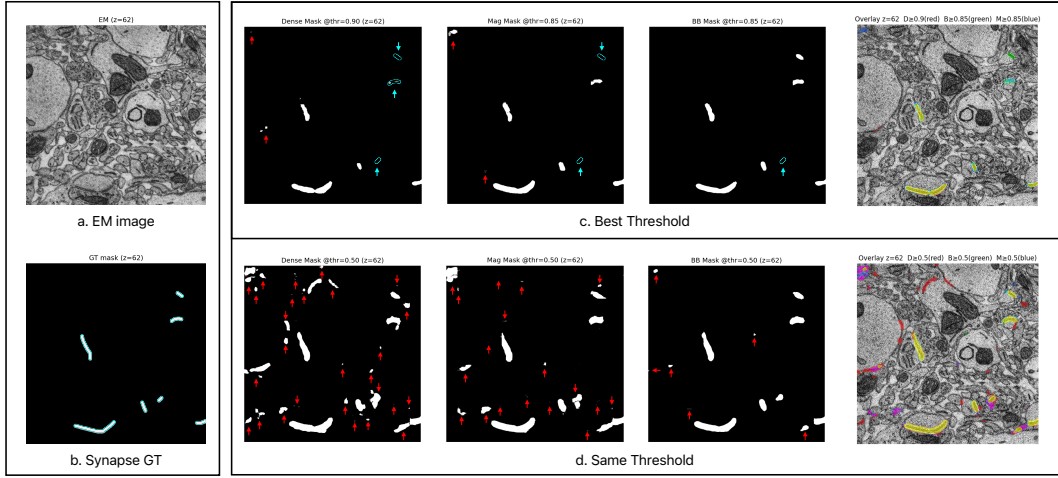

Figure 21: Synapse prediction at a shared decision threshold vs method-specific Best-F1 threshold (qualitative).

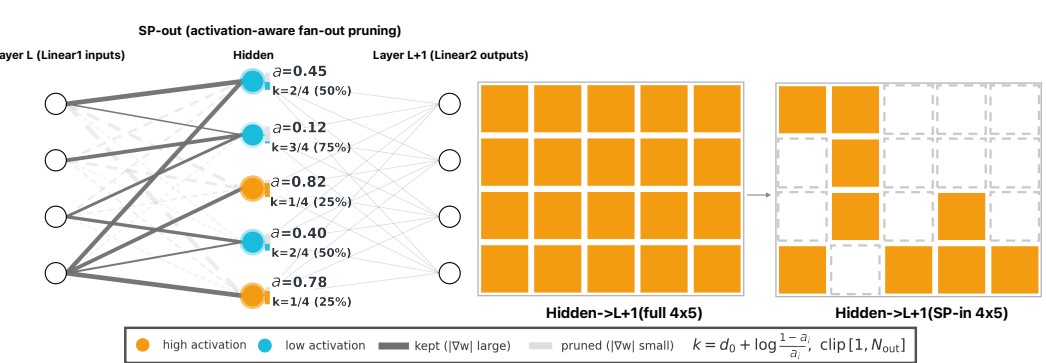

Figure 22: SP-in (dendritic pruning). Activation-aware fan-in pruning that down-regulates activity $a$.

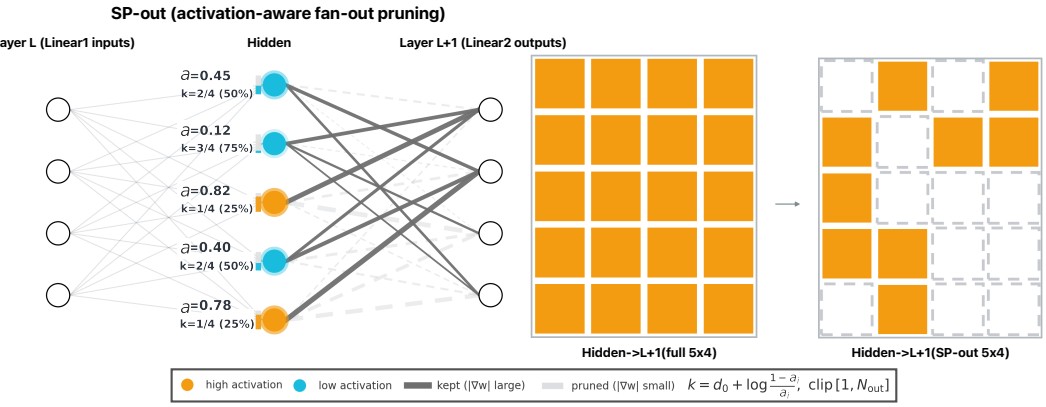

Figure 23: SP-out (axonal pruning). Activation-aware fan-out pruning that reduces audience $k$.

