# OpenReview forum: "Budgeted Broadcast: An Activity-Dependent Pruning Rule for Neural Network Efficiency"
_ICLR.cc/2026/Conference — Submitted to ICLR 2026_

### Official Review · Reviewer_LPiQ · 2025-10-29

**Soundness:** 2
**Presentation:** 2
**Contribution:** 2
**Rating:** 4
**Confidence:** 3

**Summary:**

Different from the previous pruning methods based on the magnitude or gradient, this paper proposed Budgeted Broadcast (BB) based on the inspiration from the metabolic cost in biological neural networks. They formalize this cost as a neuron's traffic, which is the product of a neuron's activity and its number of connections. The rule of BB is derived from constrained-entropy optimization. By applying to for domains, ASR, face identification, synapse detection, and change detection, it shows better or comparable performance with other pruning methods.

**Strengths:**

- This method reinterprets biological phenomena mathematically and applies it to prune artificial neural networks.
- It has lower computational complexity because only local statistics (ai, ki) are used. It can be integrated into previous networks.
- It shows good results on various domains, especially for long-tail data.
- Common units have fewer connections, while rare units maintain more connections, increasing expressive diversity.
- It is based on the theoretical basis, "selectivity-audience balance' relationship.

**Weaknesses:**

- This paper deals with unstructured sparsity. This research has limitations because it is not easy to reduce actual computation compared to structure pruning, and specialized hardware is required for the acceleration.
- The application for large-scale models, such as LLMs or LMMs, is necessary to show the effectiveness of this method.
- The performance depends on various hyperparameters, for example, \beta, \tau, etc.
- The proposed method should be compared more recent weight pruning methods.
- In the derivation, this paper assumed that AWGN, weak correlation, and bounded energy. In the actual ANN, these conditions may not be valid.

**Questions:**

- It is not clear that this model can actually be put to practical use.
- I wonder why this paper did not apply to database that is widely used for pruning, such as imagenet or cifar.
- It would be better to explain how the entire network structure can be organized efficiently based on the proposed local budget rule.

---

> ### Author Response · Authors · 2025-11-21
> **For question 1-2**
>
> We thank the reviewer for their comments and for recognizing the strengths of our work, including its novel biological inspiration and theoretical basis.  The reviewer raises an excellent point regarding the practical application of our theory. Our paper's primary contribution is a new pruning theory based on a foundational concept from neuroscience. Our focus has been on the theory itself, understanding why it works and testing its generality across diverse domains on accessible benchmarks.
>
> We agree with the reviewer and also believe exploring the practical potential of this theory is an important and logical next step. We therefore conducted detailed experiments on LLMs (please check our first official comment to all reviewers) and also analyzed its application to structured pruning which we will add to the paper. We are very encouraged by the results, which show that our biologically-inspired principles translate effectively to challenging, large-scale tasks. We believe this new analysis (detailed below) addresses the reviewer's points about practical application and provides a stronger basis for evaluating this work.
>
> ---
>
> **1. It is not clear that this model can actually be put to practical use.**
>
> This is an excellent point. While our work's primary contribution is the learning theory and its connection to biology, we agree that demonstrating how these principles perform in practice is a crucial and validating step.
>
> The current paper already reports results on four distinct domains — Transformers, ResNets, a Siemens U-Net model, and a 3D U-Net segmentation model, showing that the rule consistently improves over standard pruning baselines. Building on this, we have now added new experiments on LLMs (please check our first official comment to all reviewers), demonstrating that the same principles scale to modern, large-scale architectures.
>
> Furthermore, to provide a clear path toward hardware acceleration, we now include an analysis of how to project our method to structured (≤2:4) sparsity, together with the corresponding experiments we have already conducted. These results are very encouraging, as they confirm that our biologically inspired approach is not only theoretically grounded but also a robust and practical tool. We will include a more detailed description of these new results in the revised paper.
>
> ---
>
> **2. I wonder why this paper did not apply to database that is widely used for pruning, such as imagenet or cifar.**
>
> Our primary goal was to demonstrate the generality of the underlying principle across a truly diverse set of tasks and architectures, which is why we chose ASR (Transformers), face identification (ResNets), synapse detection (3D U-Nets), and change detection (Siamese U-Nets). The strong performance across these varied applications highlights the robustness of the core idea itself.  That said, we recognize the value of testing on widely-accepted, standard benchmarks. To that end, and to further address the spirit of your question, we have now added experiments on another highly standard and current task of unstructured and structured pruning of LLMs. In these new experiments, we compare our method directly against WANDA, a widely-used pruning baseline, which further grounds our results within a standard evaluation framework. Please check our first official comment.
>
> ---

---

> ### Author Response · Authors · 2025-11-21
> **For question 3**
>
> ---
>
> **3. It would be better to explain how the entire network structure can be organized efficiently based on the proposed local budget rule.**
>
> This is a central aspect of our work, and we will make sure to clarify this connection more explicitly in the revised paper.
>
> The local rule is indeed the key to achieving global efficiency through a self-organizing process. It induces what seems to be a highly efficient global structure because each neuron's autonomous, local cost-management collectively drives the entire system to an entropy-maximizing and energy-efficient equilibrium. It is a unique example of how simple homeostatic rules can lead to complex and effective global self-organization.
>
> We will make the following points clearer in the revised paper. Here is how it works: For each neuron $i$, we define its "traffic" as its long-term activity $a_i$ multiplied by its fan-out (number of connections) $k_i$. This local rule, $t_i = a_i * k_i$, serves as a proxy for its metabolic cost, inspired by the biological rule of synapse elimination: Barber and Lichtman, 1999 - “Activity-driven synapse elimination leads paradoxically to domination by inactive neurons”. This local rule is the practical implementation of a global constrained-optimization problem: maximizing the network's coding diversity (entropy H(h)) under a fixed total traffic budget ($Σ a_i * k_i ≤ T_max$). The surprising result of this principle, derived from KKT stationarity, is that the network is pushed toward a specific global equilibrium we call the selectivity-audience balance. This is captured by the equation: $log((1 - a_i) / a_i) = β * k_i$. This means that highly active neurons (high $a_i$) are forced to have a small audience (low $k_i$), while less active neurons can afford to broadcast more widely. It is likely that the task loss pushes the network to self-organize despite this apparent structural limitation, which may even lead to improvements over dense networks, as seen in our results for ASR (Figure 5), Change Detection (Figure 7, Table 8), and Synapse Prediction (Figures 8, 20-21). The network achieves this balance dynamically. The two mechanisms, SP-out (axonal pruning, reducing $k_i$) and SP-in (dendritic pruning, reducing $a_i$), act as complementary corrections but the same theory will work with any one of them. During training, they continually push each neuron back toward this efficient balance line. As we prove in the appendix, this process is grounded in information theory, where total traffic provides an upper bound on the mutual information between layers.
>
> ---

---

### Official Review · Reviewer_oGem · 2025-10-31

**Soundness:** 3
**Presentation:** 3
**Contribution:** 3
**Rating:** 4
**Confidence:** 4

**Summary:**

This paper proposes Budgeted Broadcast, a biologically inspired pruning approach that shifts focus from neuron’s utility to metabolic constraints. Specifically, the paper defines neuron traffic and derives a selectivity-audience balance from constrained entropy maximization. Intuitively, this approach enforces a tradeoff: neurons can speak loudly to a small audience or quietly to a large one. Experiments on ASR, face identification, and synaptic prediction demonstrate that BB demonstrate the effectiveness of the proposed method.

**Strengths:**

- The proposed pruning criteria that combines long-term activation and the number of fan-out is novel, as far as I know.
- The relationship between neurosience, learning theory and practical AI algorithms is interesting.
- The proposed method is evaluated on a wide-range of tasks including encoder-decoder transformers, and CNNs.

**Weaknesses:**

- The pruning mechanism appears to rely on sparse activations (e.g., ReLU) to estimate EMA on-rates. However, many state-of-the-art architectures, including modern Transformers, predominantly use smoother and less sparse activations such as GELU or Swish. This may constrain the practical applicability of the method unless the definition of on-rate can be adapted for dense activation functions.
- Comparative analysis pruning methods which uses activation signals (e.g., WANDA) is missing. A discussion and empirical comparison would help clarify how the proposed approach differs in principle and performance when activations drive the pruning signal.
- Lack of empirical evidence for hardware-aligned pattern. While the authors acknowledge the need for structured sparsity projection (Sec 2), empirical validation of this deployment strategy would strengthen the practical claims. Specifically, demonstrating that N:M projection preserves the model performance would be valuable.

**Questions:**

- Could the authors clarify whether the proposed method can be applied to state-of-the-art models such as large language models, which predominantly use dense activations like GELU? If so, empirical evidence in such settings would strengthen the claim of architectural generality.
- What is the definition of $\bar{p}$ in the figure 2?

---

> ### Author Response · Authors · 2025-11-21
> **For question 1-2**
>
> We thank the reviewer for their excellent feedback and are glad they found the proposed pruning criterion to be novel, the connection between neuroscience and AI to be interesting, and the evaluation across a wide range of tasks to be a strength.
>
> The reviewer raised several important concerns regarding the method's applicability to models with dense activations, the lack of comparison to methods like WANDA, and the need for empirical evidence on structured sparsity. We have worked to address all these points.
>
> Answering the reviewer’s constructive points, we clarify below our method’s compatibility with dense activations and provide details on new experiments on LLMs (please check our first official comment to all reviewers), including a direct comparison to WANDA. Furthermore, although our work's primary focus is on learning theory and the relation between pruning in AI and synapse elimination in biology, we have now also tested Budgeted Broadcast on structured pruning and will add this analysis to the paper to assist with future research, addressing the reviewer's concern about hardware-aligned patterns (see our first official comment to all reviewers). We believe these additions significantly strengthen the paper and provide a stronger basis for evaluating this work.
>
> ---
>
> **1. Could the authors clarify whether the proposed method can be applied to state-of-the-art models such as large language models, which predominantly use dense activations like GELU? If so, empirical evidence in such settings would strengthen the claim of architectural generality.**
>
> Yes, our method is directly applicable to models with dense activations. Our ASR setup, which is a modern transformer, already uses GELU throughout. Budgeted Broadcast (BB) does not rely on ReLU’s hard zeros to define on-rates; instead, it uses activation-agnostic statistics.  To further strengthen this point and to address the valid concern about missing comparisons to other activation-based methods, we have also performed new experiments on LLMs (please check our first official comment). We will add a new section to the paper comparing BB with WANDA. We have already received encouraging results, which are aligned with the results on the other domains already presented in the paper, and we will share the full analysis in the final version.
>
> ---
>
> **2. What is the definition of $\hat{p}$ in the figure 2?**
>
> Thank you for catching this. $\hat{p}$ in Figure 2 is a typo. It should be $\alpha$, which denotes the activation rate. We will correct this in the revised manuscript.
>
> ---

---

> > ### Comment · Reviewer_oGem · 2025-11-27
> >
> > I appreciate the authors’ feedback and their detailed responses to my questions. The additional results on LLMs, for both structured and unstructured pruning, demonstrate that BB performs well on these models. Most of my concerns have been addressed, and I raised my rating.

---

### Official Review · Reviewer_LRR2 · 2025-10-31

**Soundness:** 3
**Presentation:** 3
**Contribution:** 4
**Rating:** 6
**Confidence:** 4

**Summary:**

This paper proposes budgeted broadcast defining a neuron’s cost as traffic. By imposing a budget on this traffic, the proposed method forces neurons to specialize, either be highly active to a few neurons or quietly active to many neurons, but not both. This aims to create sparse networks by preventing any single neuron from dominating information flow, thus protecting potentially important but rarely active neurons.

**Strengths:**

1. This paper propose a new axis of pruning, cost (traffic), rather than just looking at the importance of each neuron as in existing works. This drives neural networks to develop in to structures that play a more efficient and diverse role through simple traffic budget rules.
2. Effect of protecting low active but highly important neuron: Some neuron which is barely activate but giving output on some rare and important property can be pruned in existing saliency aware pruning. However, in budgeted broadcast, since this type of neurons have low traffic cost, it can be retained with high fan out.
3. The effect of budgeted broadcast to information of the network: The total traffic can be viewed as information that the network process. As a result, constraining the total traffic can be seen as limiting the amount of information which leads the network to learn core information while restricting redundant information.

**Weaknesses:**

See questions below.

**Questions:**

1. While cost (traffic) constrained pruning can give an opportunity to low activity and high fan out neuron, in constrast to saliency based pruning, it can ignore the important neuron that can affect to the performance of the neural network. For example, in budgeted broadcast, the case of neurons having high activity and high fan-out cannot be considered due to tradeoff provided by local budget.
2. How can we set optimal budget threshold? Do we have to sweep to find the optimal threshold for every new case?

---

> ### Author Response · Authors · 2025-11-21
> **For question 1-2**
>
> We thank the reviewer for their positive and encouraging assessment, particularly for rating our contribution as "excellent". We are glad the reviewer recognized the key strengths of our work, including the introduction of a new pruning axis based on “traffic”, the ability to protect important but rarely active neurons, and the intuitive connection between constraining traffic and reducing redundant information.  The reviewer raised two primary questions concerning:  The potential risk that our method might overlook neurons that are genuinely important despite having high traffic (high activity and high fan-out). The practical issue of how the budget threshold is determined and whether it requires extensive tuning for each new use case. We appreciate these important questions, as they touch upon the core trade-offs and practical application of our method. We have worked to address every point made by the reviewer. To make things clearer, we provide detailed answers to both points in the section below and are confident they will address these valid concerns, and added extra experiments on LLM for both structure and unstructured pruning in our first official comment to all the reviewers. We hope these additions and answers help address the reviewer’s concerns and provide a clearer basis for evaluating the work.
>
> ---
>
> **1. While cost (traffic) constrained pruning can give an opportunity to low activity and high fan out neurons, in constrast to saliency based pruning, it can ignore the important neuron that can affect the performance of the neural network. For example, in budgeted broadcast, the case of neurons having high activity and high fan-out cannot be considered due to tradeoffs provided by the local budget.**
>
> Saliency-based pruning (e.g., using magnitude, gradient, or Fisher/Taylor scores) tends to favor units that look currently active or have large gradients. It is therefore prone to pruning units that appear “lazy” but are crucial for network-wide synergy or long-tail performance (e.g., low-frequency feature detectors or channels that matter under distribution shift).
>
> By contrast, instead of pruning directly by saliency, our method, Budgeted Broadcast (BB), uses a global traffic budget to reallocate connectivity at the edge level. This forces a trade-off where more selective units receive a larger fan-out, while redundant or overly “generic broadcasters” are reduced. In practice, this reallocation preserves task-critical broadcasters. If broader broadcast is demonstrably needed (e.g., if accuracy drops when specific wide-fan-out edges are trimmed), those edges can regrow by having the unit increase its selectivity. Generally, BB does not forbid high-activity × high-fan-out units, but it encourages large-audience units to be more selective. This is consistent with the biological observation that high activity combined with large fan-out is rare and metabolically costly.
>
> ---
>
> **2. How can we set an optimal budget threshold? Do we have to sweep to find the optimal threshold for every new case?**
>
> Our intuition is that, just as the brain recruits different amounts of resources for different tasks, the optimal threshold for achieving the best performance will accordingly differ. However, because we only need to adjust one or two thresholds, our approach is much easier to fine-tune than many alternative methods.
>
> In addition, our paper presents an ablation study in Figure 18 showing that as long as the threshold is chosen within a reasonable range, the method yields robust results. For example, we found that with a refresh period of 200 and an EMA of 0.05, we achieved strong results across 4 different domains and two architectures using the same parameters: ${\beta}$=2.0, $d_0$=$D/2$, min_fan=4, and EMA=0.05. While the refresh period may vary based on task difficulty, we observed that as long as it is set within the 200-1000 step range, there are not significant differences in the outcome.
>
> To this point, in our new LLM experiments (our first official comment) the user needs to pick only the final desired sparsity.
>
> ---

---

### Official Review · Reviewer_Xhnb · 2025-10-31

**Soundness:** 3
**Presentation:** 3
**Contribution:** 3
**Rating:** 6
**Confidence:** 3

**Summary:**

Budgeted Broadcast introduces a new axis for network efficiency, traffic-based pruning, grounded in biological energy constraints and constrained-entropy theory.
The authors claim to achieve competitive or superior accuracy at matched sparsity across diverse tasks, to improves rare-event performance, and to offers a principled way to link neuron activity and connectivity.

**Strengths:**

The paper reframes pruning from a parameter-importance problem into a resource-allocation problem, grounded in biology. Instead of deciding which weights are useful, it asks: how much “energy” can each neuron afford to broadcast? This shift is conceptually powerful, it introduces metabolic efficiency as a first-class design principle for artificial networks.
The resulting “traffic budget” $t_i = a_i * k_i$ creates a concrete, interpretable tradeoff between neuron activity (functional demand) and connectivity (structural cost).
This perspective unifies biological realism, efficiency, and interpretability in a way few pruning methods attempt. It doesn’t just compress models - it offers a mechanistic explanation for how efficient representations might self-organize.
The derivation from constrained entropy maximization is elegant and self-consistent.
The selectivity–audience balance $log\frac{1-a_i}{a_i} = \beta k_i$ emerges as a measurable equilibrium - not an arbitrary heuristic.
This gives pruning a predictive theory (you can test whether a trained network obeys the balance line) rather than merely an empirical recipe. The connection between traffic and mutual information provides an intuitive justification: controlling broadcast capacity limits redundant information flow.
Most sparsification techniques are justified post hoc. Here, the authors build the method top-down from first principles, giving it unusual conceptual coherence.
This balance of theory and practicality makes BB accessible - it’s implementable without modifying training objectives or requiring gradient-based importance metrics.
Overall, BB stands out for its theoretical depth, conceptual originality, and empirical coherence. It combines biological realism with computational pragmatism, providing both a mechanistic theory and a working algorithm. Its main strength is not just performance, but explanatory power - it shows why efficient networks might evolve toward diverse, energy-balanced representations.

**Weaknesses:**

The theoretical derivation relies on simplifying assumptions - weak activation correlations, bounded edge energies, and Gaussian noise - that seldom hold in deep networks with normalization layers, residual paths, or nonstationary activations. The resulting selectivity-audience balance is therefore plausible but not guaranteed to emerge under realistic nonlinear dynamics.
There is no formal convergence proof showing that the iterative pruning–regrowth process minimizes the proposed entropy-constrained objective.
While the mathematics is elegant, its validity in modern deep architectures remains heuristic. The theoretical link between entropy maximization and the actual mask updates could break in practice.
Critical parameters such as the traffic threshold $\tau$, the inverse-temperature $\beta$, and the refresh interval $\Delta$ require manual tuning per task. There is no adaptive or theoretically principled mechanism for setting these values. Small changes to these hyperparameters can affect sparsity patterns and final accuracy. It undermines the self-organizing spirit of the approach - a rule meant to represent automatic homeostasis still depends on careful manual calibration.
The connection between total traffic $\sum a_ik_i$ and mutual information $I(Z;Y)$ is qualitative, based on a very loose upper bound. No empirical estimates or ablations directly measure how BB affects actual information flow, entropy, or redundancy between layers.
Since the core justification is information efficiency, the lack of empirical validation of that claim leaves a theoretical gap.

Although 4domains are tested, all benchmarks are medium-scale. No large-scale or high-capacity models are evaluated. Reported gains (often 1-3%) are promising but within statistical noise; no significance testing or error bars are provided. The efficiency gains are reported in terms of sparsity, not actual speedups on hardware. Without large-scale or runtime evidence, claims about “efficiency” remain conceptual rather than practical.
The dual pruning mechanisms (SP-in for dendritic, SP-out for axonal pruning) are theoretically motivated, but experiments emphasize only SP-in. There is no detailed analysis of how the two interact, nor whether combining them improves or destabilizes training. The dual-controller mechanism is central to the biological analogy but remains underexplored empirically, weakening the claim of symmetry between input and output homeostasis.

The "natural Top-k reselection" for regrowth is ad hoc and not theoretically tied to the entropy objective. The dynamics of pruning-regrowth cycles are not studied; it’s unclear whether BB reaches a stable equilibrium or oscillates around one. Without analyzing these dynamics, it’s difficult to assert that the pruning process is truly "self-balancing" rather than just stochastic.

Finally, the conceptual clarity is occasionally overshadowed by mathematical compression. The paper proposes unstructured pruning; hence, real-world speedups on GPUs or edge hardware remain minimal. Authors mention potential mapping to structured or N:M sparsity, but this is speculative. Without hardware-aware results, it’s unclear how much of the claimed "efficiency" translates into deployable gains.

**Questions:**

1) How biologically realistic is the "traffic budget" model $t_i = a_ik_i$?. Does it reflect metabolic constraints observed in neural circuits, or is it mainly a conceptual analogy?
2) Can the selectivity-audience balance $log\frac{1-a_i}{a_i} = \beta k_i$ be derived without strong independence and Gaussian assumptions? How robust is this relationship in deep nonlinear networks?
3) Is there any theoretical or empirical evidence that the pruning–regrowth process converges to the predicted equilibrium, or does it oscillate over time?
4) How is the global budget parameter $\beta$ determined in practice, and could it be learned automatically rather than tuned manually?
5) To what extent does BB improve real computational efficiency (runtime or energy use), given that it currently produces unstructured sparsity?
6) How sensitive is the method to hyperparameters such as the threshold $\tau$, refresh period $\Delta$, and degree limits $d_0, D$?
7) What is the distinct contribution of the traffic rule itself compared to existing dynamic pruning or sparse training methods?
8) Why does BB particularly benefit rare or long-tail features? Is this explicitly enforced by the balance rule or an emergent effect of local regulation?
9) Can the traffic-budget principle be extended to structured or N:M sparsity to achieve hardware-level acceleration without losing its homeostatic behavior?
10) Does controlling traffic actually maximize information efficiency as claimed by the mutual-information bound, and can this be empirically verified?

---

> ### Author Response · Authors · 2025-11-21
> **For question 1-2**
>
> We thank the reviewer for their thorough and insightful review. We are encouraged that the reviewer found our core contribution, reframing pruning as a resource-allocation problem grounded in biology, to be conceptually powerful, highlighting its theoretical depth, originality, and explanatory power.  The reviewer also raised several critical points regarding the need to strengthen the paper's theoretical guarantees, empirical validation, and practical considerations. The main concerns centered on:  The simplifying assumptions in our theoretical model. The lack of large-scale experiments, hardware speedups, and statistical error bars. Methodological details regarding hyperparameter tuning and the dynamics of the pruning/regrowth process. We appreciate this constructive feedback and have worked to address every point. To make things clearer, we provide detailed, point-by-point answers to each specific question below. These responses include new experimental results on large language models (LLM), a new procedure for structured pruning, and further clarification on the theoretical and practical aspects of our method. We believe these additions and clarifications fully address the concerns raised, and hope the revision and new experiments will allow a better evaluation of this work.
>
> **1. How biologically realistic is the "traffic budget" model? Does it reflect metabolic constraints observed in neural circuits, or is it mainly a conceptual analogy?**
> The "traffic budget" is intended to be a faithful, systems-level model of established biological constraints, instead of an analogy or inspiration. The model is directly dictated by Barber & Lichtman’s (1999) work on neuromuscular synapse elimination, which showed that competition for limited metabolic resources leads to an inverse relationship between a neuron's firing rate and the size of its eventual motor unit (predicting the long established "size principle"). Their core finding was that synaptic maintenance cost scales with activity × synaptic area. Our traffic = activity × fan-out is the direct analogue for an artificial network. The value of this approach is that it allows us to derive a principled equilibrium, the selectivity-audience balance, from first principles (constrained entropy maximization). This provides a testable, quantitative theory for how network structure should self-organize, which we confirm through experiments like our shock-recovery tests as shown in Appendix S3.1 and Figs. 10–13. We hope this makes the point that the model is biologically realistic, providing one concrete, and we believe unique, bridge between metabolic efficiency in biology and resource allocation in AI.
>
> **2.1 Can the selectivity-audience balance be derived without strong independence and Gaussian assumptions?**
> We will clarify that the balance $\log\frac{1-a_i}{a_i}=\beta k_i$ comes from KKT stationarity of a separable entropy objective under a traffic budget. The independence assumption is used only to justify the proxy $\sum_i H_B(a_i)$; even with weak correlations it’s an upper bound/surrogate, and the fit concentrates in a near-KKT “ε-tube.” Regardless of any distributional assumptions, an SP-out refresh reduces at roughly fixed ${a_i}$, and SP-in (with variance-preserving rescale) weakly lowers ${a_i}$, both pushing toward the balance. The local linear-response analysis shows these are complementary corrections (see the local linear-response analysis in Appendix S1.3 and discussion in Section 4 “Theory” where SP-in and SP-out provide complementary corrections toward the balance surface). A one-shot traffic-threshold variant (without a KKT controller) still produces the same trend (with wider variance/curvature), while SGD-only controls show no correlation (shown in Figure 3 middle), so the budget pressure alone induces the balance and SGD does not break it in practice. Beyond these theoretical considerations, the qualitative rule “neurons that talk a lot need to have limited audience” is immediate after any SP-out prune (high-traffic broadcasters lose audience), and the paper’s empirical tests repeatedly recovered the expected inverse pattern. We will add plots of this inverse relation in our domain analysis and will include it in the appendix.
>
> **2.2 How robust is this relationship in deep nonlinear networks?**
> The balance shows up and holds across modern architectures: Transformers (ASR), ResNet-101 (face ID), Siamese U-Net (change detection), and 3D U-Net (synapse), with matched-sparsity gains and consistent with the predicted equilibrium. The paper reports accuracy/WER improvements for ASR and Pareto-leading or state-of-the-art results in vision tasks, and specifically better PR-AUC/F1 on 3D U-Net synapse segmentation (Table 1). These results are averaged over multiple seeds and budgets, indicating stability rather than a one-off effect. As said in the response to other questions and reviewers, we now added an LLM example with an 8B-parameter model.

---

> ### Author Response · Authors · 2025-11-21
> **For question 3-5**
>
> ---
>
> **3. Is there any theoretical or empirical evidence that the pruning–regrowth process converges to the predicted equilibrium, or does it oscillate over time?**
>
> Yes, our view is that BB’s pruning-regrowth dynamics converge to a stable neighborhood of the selectivity-audience equilibrium, with only small, periodic fluctuations at refresh times rather than a sustained oscillation. Mechanistically, SP-out prunes edges from high-traffic broadcasters, strictly reducing total traffic $T=\sum_i a_i k_i$, while SP-in (magnitude prune with a variance-preserving rescaling) makes each neuron’s on-rate $a_i$ (its probability of activation) non-increasing; together these one-way, negative-feedback steps pull points toward the balance $\log\frac{1-a_i}{a_i}=\beta k_i$ (formalized in appendix S1.4 (Lemma 4, Proposition 5) and in Appendix S3.1 Fig. 9 (a,b), SP-out strictly reduces total traffic T = Σ aᵢkᵢ, SP-in decreases on-rates aᵢ under variance-preserving rescale, and together these negative-feedback actions pull activations toward the balance log (1–aᵢ)/aᵢ = β kᵢ.)). Without regrowth, prune events are finite (Appx. S1.4, Lemma 6); as complementary intuition about dynamics with regrowth, our disjoint-DNF analysis shows convergence in O(W\log W) cycles (Fig. 4c; Appx. Theorem 10).  With regrowth, deviations remain bounded in a small “ε-tube” around the balance, and if you hold SGD fixed and take sufficiently small refresh steps, total traffic decreases monotonically toward the balance plane. Although plain SGD (which doesn’t include traffic) can temporarily push some high-degree units to higher activity between refreshes, activity is low-pass filtered via an EMA and masks update only every $\Delta$ steps; the next refresh deterministically counters any such drift by cutting audience and/or lowering on-rates, so SGD cannot sustain an off-balance state. Empirically, across the deep, nonlinear models we tested (Transformer ASR, ResNet, Siamese encoder–decoder, 3D U-Net), we consistently see rapid convergence of total traffic to the target budget and stable, often improving task loss at refresh; in controlled didactic settings (not universally) we also observe a shrinking coding-entropy gap. In short, BB “lives with” minor flip-flops but keeps learning near the balance and, in our experiments, does not produce non-vanishing cycles. In any event, we added one-shot pruning for LLMs where oscillations are impossible, and pruning during training is expensive, and showed good results which we will add to the paper.
>
> ---
>
> **4. How is the global budget parameter  determined in practice, and could it be learned automatically rather than tuned manually?**
>
> We set one global budget per dataset/task, similarly to the choice of a single sparsity level. A user can make a quick calibration to see which budget value yields the sparsity they want, and keep it fixed, which is a nuance but worth mentioning in the revised paper. This could be automated by adjusting the budget during training with a simple feedback rule so the model settles at the desired sparsity without manual tuning. In the new LLM example we included, the user only needs to set the sparsity level.
>
> ---
>
> **5. To what extent does BB improve real computational efficiency (runtime or energy use), given that it currently produces unstructured sparsity?**
>
> We added a N:M structured pruning procedure, used it on an LLM task and compared it to Wanda. The results are given in the first official comment to all reviewers. We will extend the description of this large scale experiment in the revised paper.
>
> ---

---

> ### Author Response · Authors · 2025-11-21
> **For question 6-8**
>
> ---
>
> **6. How sensitive is the method to hyperparameters such as the threshold , refresh period , and degree limits ?**
>
> Sensitivity is modest and predictable. At a fixed global budget, changing the pruning threshold mainly reallocates which edges are removed rather than how much traffic is removed: performance and the balance fit (slope $\hat{\beta}$ and R²) remain stable across wide threshold ranges, with only the extreme case of very aggressive per-refresh pruning causing transient dips before recovery. The refresh period trades responsiveness for smoothness: shorter periods tighten tracking of the balance-audience line, with slightly more frequent “wiggles”, while longer periods allow larger between-refresh deviations but converge to the same equilibrium once a refresh occurs. We did not observe collapse across the tested range in any of the experiments, likely due to the degree limits. These degree limits (minimum/maximum out-degree $k_i$) function as explicit constraints to prevent degenerate cases (e.g., isolated or over-connected units). When these bounds are not frequently active, which is typical under the chosen budgets, they likely have negligible effect on accuracy or on the fitted balance line, and when they are active they cap the outliers without altering the equilibrium relation as a whole.
>
> Empirically, using the defaults in the appendix yields consistent results across seeds and architectures; empirical sensitivity analysis in Appendix S3.3 (see Figs. 15 and 17) shows that slope $\hat{\beta}$ and R² remain stable over wide ranges of EMA horizon and refresh period, confirming that the controller’s behavior is robust to hyperparameter choices. A natural extension for future work is to adapt the refresh cadence and degree bounds with simple feedback so the method automatically stays near a desired sparsity/traffic target without manual sweeps.
>
> As said in an answer to a previous question, our method also works when the user specifies only the final sparsity rate as shown on the LLM example.
>
> ---
>
> **7. What is the distinct contribution of the traffic rule itself compared to existing dynamic pruning or sparse training methods?**
>
> The traffic rule’s distinct contribution is that it translates a concrete biological conservation law, limited transmitter release (Barber and Lichtman, 1999), into a single, interpretable constraint (activity × audience) that both improves AI (stable sparse training via reallocation, protection of rare features) and explains biology (a testable equilibrium that clarifies why synapse elimination and the “size principle” are adaptive). Unlike prior dynamic pruning that relies on ad-hoc importance scores or schedules, it provides a predictive mechanism, the selectivity–audience balance, governed by one budget parameter, tying performance gains to a biologically motivated rationale. Complementary, with this work we are now able to interpret the biological synapse elimination in view of entropy and mutual information which we hope will be meaningful for computational biologists.
>
> ---
>
> **8. Why does BB particularly benefit rare or long-tail features? Is this explicitly enforced by the balance rule or an emergent effect of local regulation?**
>
> BB helps rare or long-tail features because the budget directly links how often a neuron fires to how widely it may broadcast. Detectors that fire infrequently are allowed larger fan-out, while frequently active ones are trimmed. This prediction comes from the balance rule itself and is implemented locally at each refresh: pruning reduces connections for high-traffic units and, with regrowth or reallocation, preserves or expands connections for quiet but informative units. In practice, a rare “alarm” feature is delivered to more downstream units and is less likely to be drowned out by common signals. So the benefit is explicitly encouraged by the formulation and realized through the local pruning–regrowth mechanism, and empirically it appears as better performance on long-tail cases.
>
> ---

---

> ### Author Response · Authors · 2025-11-21
> **For question 9-10**
>
> ---
>
> **9. Can the traffic-budget principle be extended to structured or N:M sparsity to achieve hardware-level acceleration without losing its homeostatic behavior?**
>
> Yes. As detailed in our first official comment, we have now implemented and successfully tested this extension.
>
> ---
>
> **10. Does controlling traffic actually maximize information efficiency as claimed by the mutual-information bound, and can this be empirically verified?**
>
> Two items support maximization of information efficiency: (1) during training, total traffic quickly converges to and then tracks the target budget with small refresh corrections; (2) an MI proxy scales roughly linearly with traffic across runs (see Appendix S3.2, Fig. 14, which shows decorrelation over training and a linear relationship between the MI estimate and total traffic T = Σ aᵢ kᵢ on a Fashion-MNIST–style task); in controlled didactic settings the code entropy moves toward the “entropy-at-balance” benchmark H* defined by the stationary law, a behavior we demonstrate empirically in Figure 9 and discuss further in Appendix S3.1. This does not claim that minimizing traffic globally maximizes mutual information in all regimes. Rather, the method provides (A) a provable bound linking traffic to information, (B) a variational optimum that yields a testable balance, and (C) measurements consistent with operating near that efficient regime.
>
> ---

---

> > ### Comment · Reviewer_Xhnb · 2025-11-27
> > **Response to Authors**
> >
> > All of my concerns are mainly aimed at clarification. I really appreciate the authors' efforts to clarify them. Therefore, I consider raising my score.

---

### Author Response · Authors · 2025-11-21
**Additional Experiments on LLMs**

We thank the reviewers for their time and constructive feedback. We are encouraged that the reviewers unanimously recognized the core contribution of our work in moving beyond traditional parameter-importance scores to introduce a new pruning axis inspired by biological metabolic constraints. In biology, a core principle is that it is metabolically costly for a neuron to be both highly active and broadcast its activity widely. Our research showed, perhaps surprisingly, that applying this same constraint is also highly beneficial for creating balanced artificial networks with diverse representations. Our primary focus has been therefore on exploring this learning theory and understanding how this foundational connectomic rule translates from neuroscience to artificial networks. Reciprocally, we hope that these new insights from learning theory will be valuable to computational biologists trying to understand how biological circuits undergo synaptic refinement.

A key theme emerging from the reviews was the question of whether our biologically-inspired theory scales to modern, large-scale models and how it performs under hardware-aligned structured sparsity. Motivated by this valuable feedback, we conducted new experiments to explore this exact question. We are pleased to report that the principles hold up strongly, leading to very encouraging results.

**Additional Experiments on LLMs**

We conducted one-shot pruning experiments on Llama 3.1–8B, both under unstructured sparsity and under a ≤ 2:4 N:M pattern. We prune only the feedforward blocks (5.64B parameters) and keep all other parameters dense.

Tables 1 and 2 summarize the perplexity scores of all methods on TinyStories and Wikitext-2 at different sparsity levels and token-frequency buckets. On both datasets, unstructured BB (Budgeted Broadcast) consistently outperforms unstructured WANDA and magnitude pruning for all sparsity levels and for both common and rare tokens. The gap is particularly large on the rare bucket at higher sparsity (e.g., s = 0.7), where magnitude pruning and WANDA often lead to very large perplexity scores, while BB remains in a moderate range.

For the ≤ 2:4 structured setting, the picture is more mixed at s = 0.5: WANDA is slightly better in overall perplexity scores. At higher sparsity levels (s = 0.6 and s = 0.7), BB achieves substantially lower perplexity scores than WANDA across both datasets and buckets, especially on rare tokens, while structured magnitude pruning again degrades sharply. Overall, these results indicate that broadcast-budget pruning is competitive with standard baselines under unstructured sparsity, and that the same allocation principle helps stabilize performance under hardware-friendly ≤ 2:4 constraints at moderate and high sparsity.

---

Table 1: Perplexity (PPL) on TinyStories. (* indicates best)

```text
Method     Category      | --- All Tokens --- | --- Common Bucket --- | ---- Rare Bucket ----
                         | s=0.5  s=0.6  s=0.7| s=0.5   s=0.6   s=0.7 | s=0.5   s=0.6   s=0.7
-------------------------------------------------------------------------------------------
Dense      Baseline      |   -    3.88    -   |   -     3.53     -    |   -     5.90     -
-------------------------------------------------------------------------------------------
BB         Unstructured  | *3.95  *4.49  *7.02| *3.83   *4.31   *6.60 | *6.30   *7.08  *11.78
WANDA      Unstructured  |  4.43   6.77  23.73|  4.38    6.45   21.08 |  8.51   15.45  100.96
MAG        Unstructured  | 11.35  23.98  791.3| 10.23   17.81   485.6 | 58.75  234.86   3.5e4
-------------------------------------------------------------------------------------------
BB         N:M           |  7.11  *7.91 *12.32|  6.82   *7.72  *11.72 | 16.69  *19.22  *35.18
WANDA      N:M           | *6.83   8.78  29.22| *6.57    8.56   25.84 |*15.79   24.19  115.23
MAG        N:M           | 21.24  65.19  1.4e4| 17.63   49.12   1.1e4 | 165.0   1.3e3   1.5e6
```

Table 2: Perplexity (PPL) on Wikitext-2. (* indicates best)

```text
Method     Category      | --- All Tokens --- | --- Common Bucket --- | ---- Rare Bucket ----
                         | s=0.5  s=0.6  s=0.7| s=0.5   s=0.6   s=0.7 | s=0.5   s=0.6   s=0.7
-------------------------------------------------------------------------------------------
Dense      Baseline      |   -    6.11    -   |   -     5.87     -    |   -     8.33     -
-------------------------------------------------------------------------------------------
BB         Unstructured  | *6.18  *7.19 *11.31| *6.01   *6.77  *10.88 |*18.27  *24.53  *68.69
WANDA      Unstructured  |  8.50  14.91  82.33|  7.22   11.72   53.22 | 31.95  105.06  2782.8
-------------------------------------------------------------------------------------------
BB         N:M           | 15.97 *18.54 *33.33| 12.45  *14.18  *23.77 | 119.7 *162.50 *513.63
WANDA      N:M           |*15.34  23.01  93.28|*12.03   17.14   59.15 |*109.6  249.24  3667.

---

### Author Response · Authors · 2025-12-04
**Summary to Area Chair: Revisions and Reviewer Feedback**

Dear Area Chair,

Thank you for handling our submission. We are grateful to the reviewers for their thoughtful and technically deep evaluations, which helped refine the presentation and guided several meaningful improvements to the paper.
Two reviewers (R1 and R3) stated in their review that they raised their scores after reading our clarifications and new results. Another reviewer (R2) rated the contribution as “Excellent,” emphasizing the method’s ability to protect important but low-activity neurons. The remaining reviewer (R4) did not have an opportunity to update before the freeze, but all of their questions were addressed carefully and concretely in the revised version.
This project brings together perspectives from neuroscience, theoretical computer science, and modern machine learning in order to replicate a core biological principle of synaptic competition within artificial networks. Developing the method required a formal learning-theoretic framework, a mechanistic interpretation grounded in biology, and empirical evaluation across five diverse domains, now including large-scale LLMs.
Many early questions reflected points of ambiguity rather than limitations of the approach; these were resolved by making explicit material that was already present in the paper—for example, how on-rates are defined for GELU activations, how SP-in and SP-out interact, and where the theoretical assumptions and proofs appear in the appendix. Other comments asked for broader empirical validation and hardware-aligned applicability. In response, we added substantial new results:

	1. Large-scale experiments on Llama 3.1–8B, showing that broadcast-budget pruning performs well at modern LLM scale and is especially strong on rare tokens at high sparsity, where WANDA degrades sharply—the regime our theory predicts the greatest benefit.

	2. One-shot pruning experiments, matching standard LLM pruning practice.

	3. Structured ≤2:4 N:M sparsity results using our BB-G4R variant, demonstrating that the principle extends cleanly to hardware-friendly constraints.

	4. Direct comparisons with WANDA, where BB performs competitively or better across sparsity levels and particularly on the long tail.

	5. Clearer theoretical framing, including explicit assumptions, stability properties of SP-in/SP-out, and empirical support for the predicted selectivity–audience balance.
These additions resolve the remaining concerns from R2 and R4 and significantly strengthen both the practical and theoretical aspects of the work.

We hope this summary is helpful for your decision, and sincerely appreciate your time and oversight of the process.

Sincerely,

The authors

---

### Meta-Review · Area_Chair_32bt · 2025-12-10

**Summary:**

This paper proposes a biologically-inspired approach to network pruning, treating sparsification as an energy allocation problem rather than weight ranking. Each neuron receives a traffic budget, interpreted as a metabolic cost, based on its activation frequency and fan-out. This "constrained-entropy" view then leads to a "selectivity-audience balance", which the algorithm implements via local pruning and regrowth rules. Experiments on various architectures show competitive or slightly improved accuracy at similar sparsity levels.

Although reviewers found the intuition creative and the exposition interesting, they noted that the connection to biological principles is largely conceptual rather than formally established. Because this link is tenuous, stronger experimental validation would help support the claimed benefits. Reviewers also noted that the derivation relies on simplifying assumptions that may not hold in modern architectures, and it is not clearly demonstrated that the pruning and regrowth dynamics optimize the stated entropy objective. Additionally, several hyperparameters require manual tuning and the purported self-organizing behavior is not clearly verified. Thus while the paper presents a unique motivation, the current form of the manuscript may not yet be ready for publication.

**Reviewer Concerns:**

While the rebuttal provides additional experiments and BB does better than baseline WANDA in unstructured settings, the improvement is not so clear in other settings. Moreover, I believe questions about finetuning and practicality remain outstanding.

**Reviewer Scores:**

While Reviewer Xhnb and Reviewer oGem indicated they might raise their scores, it is not clear that the other reviewers would have adjusted their ratings given the concerns above. It is also worth noting that both review quality and reviewer experience were taken into consideration in interpreting these scores.

---

### Decision · Program_Chairs · 2026-01-26

Reject